# DIFFUSION-TS: INTERPRETABLE DIFFUSION FOR GENERAL TIME SERIES GENERATION

**Xinyu Yuan, Yan Qiao**[*]
Hefei University of Technology
`yxy5315@gmail.com, qiaoyan@hfut.edu.cn`

## ABSTRACT

Denoising diffusion probabilistic models (DDPMs) are becoming the leading paradigm for generative models. It has recently shown breakthroughs in audio synthesis, time series imputation and forecasting. In this paper, we propose Diffusion-TS, a novel diffusion-based framework that generates multivariate time series samples of high quality by using an encoder-decoder transformer with disentangled temporal representations, in which the decomposition technique guides Diffusion-TS to capture the semantic meaning of time series while transformers mine detailed sequential information from the noisy model input. Different from existing diffusion-based approaches, we train the model to directly reconstruct the sample instead of the noise in each diffusion step, combining a Fourier-based loss term. Diffusion-TS is expected to generate time series satisfying both interpretablity and realness. In addition, it is shown that the proposed Diffusion-TS can be easily extended to conditional generation tasks, such as forecasting and imputation, without any model changes. This also motivates us to further explore the performance of Diffusion-TS under irregular settings. Finally, through qualitative and quantitative experiments, results show that Diffusion-TS achieves the state-of-the-art results on various realistic analyses of time series.

## 1 INTRODUCTION

Time series is ubiquitous in real-world problems, playing a crucial component in a wide variety of domains such as finance, medicine, biology, retail, and climate modeling (Lim & Zohren, 2021). However, lack of access to these dynamical data is a key hindrance to the development of machine learning solutions in some cases where data sharing may lead to privacy breaches (Alaa et al., 2021). Synthesizing realistic time series data is viewed as a promising solution and has received increasing attention driven by advances in deep learning. With perceptual qualities superior to GANs while avoiding the optimization challenges of adversarial training, score-based diffusion models (Song et al., 2021; 2020), especially denoising diffusion probabilistic models (DDPMs) (Ho et al., 2020), have taken the world of image, video, and text generation (Ho et al., 2022; Li et al., 2022a; Dhariwal & Nichol, 2021; Harvey et al., 2022) by storm than ever before.

It is hopeful that the diffusion models can be extended to the time-series area to tackle the challenging problem of high-quality time series generation. Although some recent works pioneered the extension of diffusion models to time-series-related applications, almost all of them are designed for task-agnostic generation (e.g., imputation (Tashiro et al., 2021; Alcaraz & Strodthoff, 2022) and forecasting (Li et al., 2022b; Shen & Kwok, 2023)) that train and sample with additional information. Meanwhile, the rare work on unconditional time-related synthesis with diffusion models focus on synthesizing univariate (Kong et al., 2021; Kollovieh et al., 2023) or short time series Lim et al. (2023). But first, these diffusion-based methods (Lim et al., 2023; Das et al., 2023) typically employ Recurrent Neural Networks (RNNs) as the backbone to jointly model temporal dynamics and complex correlations. These autoregressive methods have limited long-range performance due to error accumulation and slow inference speed. The second challenge lies in that the plentiful combinations of independent components like trend, seasonality, and local idiosyncrasy of a real-world time series are usually destroyed by gradually adding the noise to data in the diffusion process. As a result,

---

[*]Corresponding author. The code is available at https://github.com/Y-debug-sys/Diffusion-TS.

they can hardly recover the lost temporal dynamics since the temporal properties have not been intentionally preserved. It becomes exacerbated when time series has apparent seasonal oscillations since existing solutions lack the inductive bias to initiatively capture periodicity (LIU et al., 2022). Furthermore, it is also difficult for them to provide expert knowledge to explain both conditional and unconditional generation, and therefore they often lack interpretability.

To better tackle the aforementioned challenges, in this paper, we aim to develop Diffusion-TS, a non-autoregressive diffusion model for synthesizing time series of high quality in various scenarios, in which we explicitly model the temporal dynamics of highly complicated (e.g., multivariate and long-term) time series by introducing a transformer-based architecture for the underlying model that learns a disentangled seasonal-trend constitution of time series. This is achieved by imposing different forms of constraints on different representations. These disentangled representations not only offer Diffusion-TS an interpretable perspective on general synthetic tasks, they also plays a role in guiding the capture of complicated periodic dependencies beyond much simplified assumptions. Additionally, we design a Fourier-based loss to reconstruct the samples rather than the noises in each diffusion step, which leads to a more accurate generation of the time series. Another notable design in Diffusion-TS is a conditional generation method called reconstruction-based sampling, which makes the Diffusion-TS versatile for various conditional applications, such as time series imputation and forecasting, leading to greater flexibility without requiring any parametrically updating.

In summary, our major contributions are as follows:

- We propose a time series generation framework named Diffusion-TS, which combines seasonal-trend decomposition techniques with denoising diffusion models. This is achieved by a Fourier-based training objective, and the embedding of a deep decomposition architecture. The framework allows the model to learn meaningful temporal properties from the data, making it a highly efficient and interpretable solution for general time series generation.

- For conditional generation, we adopt an instance-aware guidance strategy built on target metric (e.g. reconstruction), which enables Diffusion-TS to adapt different controllable generative tasks in a plug-and-play way.

- Our experiments demonstrate that Diffusion-TS can generate realistic time series while maintaining high degree of diversity and novelty under challenging settings, and is competitive with existing diffusion-based methods designed for downstream applications. We also present the explanability of the model with several case studies.

## 2 PROBLEM STATEMENT

We denote $X_{1:\tau} = (x_1, \ldots, x_\tau) \in \mathbb{R}^{\tau \times d}$ as a time series covering a period of $\tau$ time steps, where $d$ denotes the dimension of observed signals. Given the dataset $DA = \left\{ X_{1:\tau}^i \right\}_{i=1}^N$ of $N$ samples of time-series signals, our unconditional goal is to use a diffusion-based generator to approach the function of $\hat{X}_{1:\tau}^i = G(Z_i)$ which maps Gaussian vectors $Z_i = (z_1^i, \ldots, z_t^i) \in \mathbb{R}^{\tau \times d \times T}$ to the signals that are most similar with the signals in $DA$, where $T$ denotes the total diffusion step. In our method, we consider the following time series model with trend and multiple seasonality as

$$x_j = \zeta_j + \sum_{i=1}^m s_{i,j} + e_j, \quad j = 0, 1, \ldots, \tau - 1, \tag{1}$$

where $x_j$ represents the observed time series, $\zeta_j$ denotes the trend component, $s_{i,j}$ is the $i$-th seasonal component and $e_j$ denotes the remainder part which contains the noise and some outliers at time $j$. And the goal of controllable generation is to generate samples from a conditional distribution $p(.|y)$, where $y$ is a control variable can be any real-world signal that will dictate the synthesis.

## 3 DIFFUSION-TS: INTERPRETABLE DIFFUSION FOR TIME SERIES

As aforementioned, time series usually exhibit complex patterns in many real-world scenarios. Inspired by the effectiveness of seasonal-trend decomposition analysis in time series modeling, the core idea of Diffusion-TS is to introduce an interpretable decomposition architecture to the underlying network based on a transformer. We take such formulation for three main reasons: (i) the use of disentangled patterns in the diffusion model has yet to be explored; (ii) our method is highly

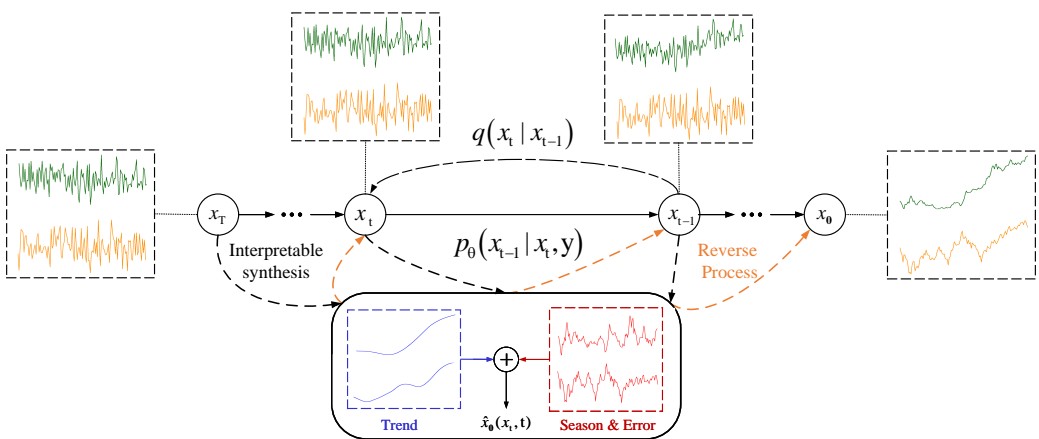

Figure 1: The forward and reverse diffusion on time series data in Diffusion-TS. The denoising network learns to predict the clean time series $\hat{x}_0$ from $x_t$ based on interpretable decomposition. In the reverse pass, the generator network gradually injects the expert knowledge to make the synthetic series target move toward the real ones.

interpretable owing to the specific designs of architecture and objective; (iii) temporal information is isolated into several parts, and it enables us to capture the potentially complex dynamics from diffused data with explainable disentangled representations. The learned disentanglement also tends to make the imputation or forecasting process more reliable (LIU et al., 2022). Thus finally, we will show how to conduct time series imputation and forecasting by the trained off-the-shelf diffusion model during the sampling process.

## 3.1 DIFFUSION FRAMEWORK

As shown in Figure 1, we start by introducing the diffusion model that typically contains two processes: forward process and reverse process. In this setting, a sample from the data distribution $x_0 \sim q(x)$ is gradually noised into a standard Gaussian noise $x_T \sim \mathcal{N}(0, \mathbf{I})$ by the forward process, where the transition is parameterized by $q(x_t|x_{t-1}) = \mathcal{N}(x_t; \sqrt{1 - \beta_t}x_{t-1}, \beta_t\mathbf{I})$ with $\beta_t \in (0, 1)$ as the amount of noise added at diffusion step $t$. Then a neural network learns the reverse process of gradually denoising the sample via reverse transition $p_\theta(x_{t-1}|x_t) = \mathcal{N}(x_{t-1}; \mu_\theta(x_t, t), \Sigma_\theta(x_t, t))$. Learning to clean $x_T$ through the reversed diffusion process can be reduced to learning to build a surrogate approximator to parameterize $\mu_\theta(x_t, t)$ for all $t$. Ho et al. (2020) trained this denoising model $\mu_\theta(x_t, t)$ using a weighted mean squared error loss which we will refer to as

$$\mathcal{L}(x_0) = \sum_{t=1}^{T} \mathop{\mathbb{E}}_{q(x_t|x_0)} \|\mu(x_t, x_0) - \mu_\theta(x_t, t)\|^2, \tag{2}$$

where $\mu(x_t, x_0)$ is the mean of the posterior $q(x_{t-1}|x_0, x_t)$. This objective can be justified as optimizing a weighted variational lower bound on the data log likelihood. Also note that the original parameterization of $\mu_\theta(x_t, t)$ can be modified in favour of $\hat{x}_0(x_t, t, \theta)$ or $\epsilon_\theta(x_t, t)$. Please refer to Appendix B for details.

## 3.2 DECOMPOSITION MODEL ARCHITECTURE

Here on a high level, we choose an encoder-decoder transformer that enhances the models' ability to capture global correlation and patterns of time series. This way, information of the entire noisy sequence is encoded before decoding. We renovate the decoder as a deep decomposition architecture as shown in Figure 2. The decoder adopts a multilayer structure in which each decoder block contains a transformer block, a feed forward network block and interpretable layers (Trend and Fourier synthetic layer). Detailed descriptions of the whole model can be found in Appendix E, we are now ready to elaborate on the details of the disentangled representation.

We achieve the disentanglement by enforcing distinct forms of constraints on different components, which introduce distinct inductive biases into these components and make them more liable to learn specific semantic knowledge. Trend representation captures the intrinsic trend which changes gradually and smoothly, and seasonality representation illustrates the periodic patterns of the signal. Error

representation characterizes the remaining parts after removing trend and periodicity. Before the start, we define the input of interpretable layers as $w_{(\cdot)}^{i,t}$, where $i \in 1, \ldots, D$ denotes the index of the corresponding decoder block at diffusion step $t$.

**Trend Synthesis.** The trend component describes the smooth underlying mean of the data, which aims to model slow-varying behavior. To produce reasonable trend components, We use the polynomial regressor (Oreshkin et al., 2020; Desai et al., 2021) to model the trend $V_{tr}^t$ as follows:

$$V_{tr}^t = \sum_{i=1}^{D} \left( \boldsymbol{C} \cdot \text{Linear}(w_{tr}^{i,t}) + \mathcal{X}_{tr}^{i,t} \right), \quad \boldsymbol{C} = [1, c, \ldots, c^p] \tag{3}$$

where $\mathcal{X}_{tr}^{i,t}$ is the mean value of the output of the $i^{th}$ decoder block, and '$\cdot$' denotes tensor multiplication. Here slow-varying poly space $\boldsymbol{C}$ is the matrix of powers of vector $c = [0, 1, 2, \ldots, \tau - 2, \tau - 1]^T / \tau$, and $p$ is a small degree (e.g. $p = 3$) to model low frequency behavior.

**Seasonality & Error Synthesis.** In this part, we will try to recover components other than trends in the model input. This includes the periodic components (Seasonality) and the non-periodic ones (Error). The main challenge is to automatically identify seasonal patterns from the noisy input $x_t$. Inspired by the trigonometric representation of seasonal components based on Fourier series (De Livera et al., 2011; Woo et al., 2022), we capture the seasonal component of the time series in Fourier synthetic layers using Fourier bases:

$$A_{i,t}^{(k)} = \left| \mathcal{F}(w_{seas}^{i,t})_k \right|, \quad \Phi_{i,t}^{(k)} = \phi \left( \mathcal{F}(w_{seas}^{i,t})_k \right), \tag{4}$$

$$\kappa_{i,t}^{(1)}, \cdots, \kappa_{i,t}^{(K)} = \underset{k \in \{1, \cdots, \lfloor \tau/2 \rfloor + 1\}}{\arg \text{TopK}} \{A_{i,t}^{(k)}\}, \tag{5}$$

$$S_{i,t} = \sum_{k=1}^{K} A_{i,t}^{\kappa_{i,t}^{(k)}} \left[ \cos(2\pi f_{\kappa_{i,t}^{(k)}} \tau c + \Phi_{i,t}^{\kappa_{i,t}^{(k)}}) + \cos(2\pi \bar{f}_{\kappa_{i,t}^{(k)}} \tau c + \bar{\Phi}_{i,t}^{\kappa_{i,t}^{(k)}}) \right], \tag{6}$$

where $\arg \text{TopK}$ is to get the top $K$ amplitudes and $K$ is a hyperparameter. $A_{i,t}^{(k)}, \Phi_{i,t}^{(k)}$ are the phase, amplitude of the $k$-th frequency after the discrete Fourier transform $\mathcal{F}$ respectively. $f_k$ represents the Fourier frequency of the corresponding index $k$, and $\bar{(\cdot)}$ denotes $(\cdot)$ of the corresponding conjugates. In fact, the Fourier synthetic layer selects bases with the most significant amplitudes in the frequency domain, and then returns to the time domain through an inverse transform to model the seasonality. Finally, we can obtain the original signal by the following equations:

$$\hat{x}_0(x_t, t, \theta) = V_{tr}^t + \sum_{i=1}^{D} S_{i,t} + R, \tag{7}$$

where $R$ is the output of the last decoder block, which can be regarded as the sum of residual periodicity and other noise.

### 3.3 FOURIER-BASED TRAINING OBJECTIVE

To enforce the model unsupervised uncover these time-series components, we train the neural network to predict an estimate $\hat{x}_0(x_t, t, \theta)$ directly. Then the reverse process shown in Figure 1 can be approximated via

$$x_{t-1} = \frac{\sqrt{\bar{\alpha}_{t-1}} \beta_t}{1 - \bar{\alpha}_t} \hat{x}_0(x_t, t, \theta) + \frac{\sqrt{\alpha_t}(1 - \bar{\alpha}_{t-1})}{1 - \bar{\alpha}_t} x_t + \frac{1 - \bar{\alpha}_{t-1}}{1 - \bar{\alpha}_t} \beta_t z_t, \tag{8}$$

where $z_t \sim \mathcal{N}(0, \mathbf{I})$, $\alpha_t = 1 - \beta_t$ and $\bar{\alpha}_t = \prod_{s=1}^{t} \alpha_s$. We take the following reweighting strategy:

$$\mathcal{L}_{simple} = \mathbb{E}_{t,x_0} \left[ w_t \| x_0 - \hat{x}_0(x_t, t, \theta) \|^2 \right], \quad w_t = \frac{\lambda \alpha_t (1 - \bar{\alpha}_t)}{\beta_t^2}, \tag{9}$$

where $\lambda$ is a constant, i.e. 0.01. Similar to Ho et al. (2020), these loss terms are down-weighted at small $t$ to force the network focus on larger diffusion step.

In addition, we propose to guide the interpretable diffusion training by applying it to the frequency domain with the Fourier transform, a mathematical operation that converts a finite-length time domain signal to its frequency domain representation (Bracewell & Bracewell, 1986). Fons et al. (2022) shows that the Fourier-based loss term is beneficial for the accurate reconstruction of the time series signals. Formally, we have

$$\mathcal{L}_\theta = \mathbb{E}_{t,x_0} \left[ w_t \left[ \lambda_1 \| x_0 - \hat{x}_0(x_t, t, \theta) \|^2 + \lambda_2 \| \mathcal{FFT}(x_0) - \mathcal{FFT}(\hat{x}_0(x_t, t, \theta)) \|^2 \right] \right], \tag{10}$$

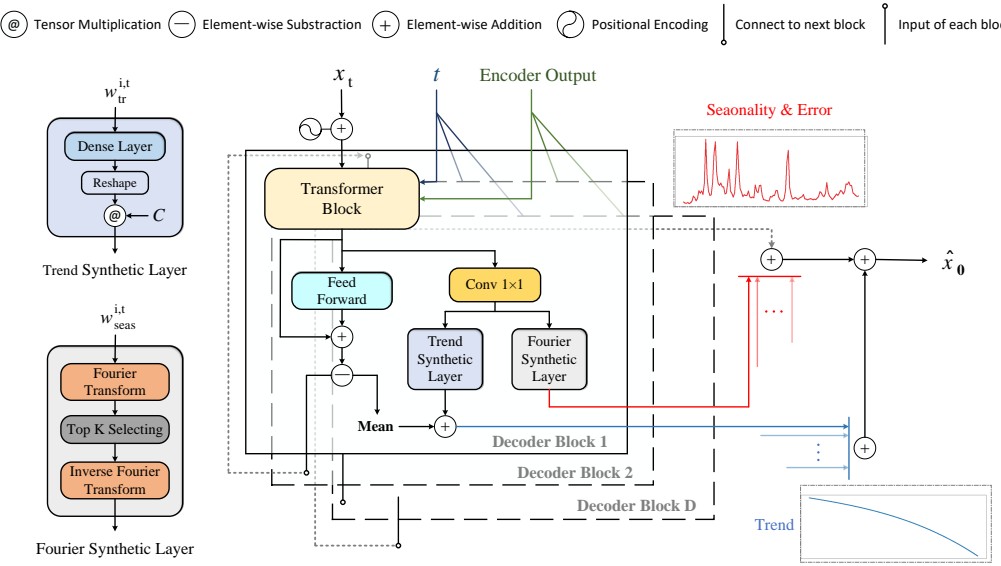

Figure 2: The network architecture of the decoder in proposed $\hat{x}_0$ approximator.

where $\mathcal{FFT}$ denotes the Fast Fourier Transformation Elliott & Rao (1982), and $\lambda_1$, $\lambda_2$ are weights to balance two losses.

### 3.4 CONDITIONAL GENERATION FOR TIME SERIES APPLICATIONS

The above has described the details of the unconditional time series generation. In this section, we will describe conditional extensions of the Diffusion-TS, in which the modeled $x_0$ is conditioned on targets $y$. The goal is to utilize a pre-trained diffusion model and the gradients of a classifier to approximately sample from the posterior $p(x_{0:T}|y) = \prod_{t=1}^{T} p(x_{t-1}|x_t, y)$, where $p(x_{t-1}|x_t, y) \propto p(x_{t-1}|x_t) \, p(y|x_{t-1}, x_t)$. Here using Bayes' theorem, we run gradient update on $x_{t-1}$ to control generation via the following score function:

$$\nabla_{x_{t-1}} \log p(x_{t-1}|x_t, y) = \nabla_{x_{t-1}} \log p(x_{t-1}|x_t) + \nabla_{x_{t-1}} \log p(y|x_{t-1}), \quad (11)$$

where $\log p(x_{t-1}|x_t)$ is defined by diffusion models, while $\log p(y|x_{t-1})$ is parametrized by an classifier, which can be approximated by $\nabla_{x_{t-1}} \log p(y|x_{0|t-1})$, proved in Chung et al. (2022).

The method can be explained as a way to guide the samples towards areas where the classifier has a high likelihood. Then given conditional part $x_a$ and generative part $x_b$, our proposed method to approximate conditional sampling for imputation and forecasting can be defined as follows:

$$\tilde{x}_0(x_t, t, \theta) = \hat{x}_0(x_t, t, \theta) + \eta \nabla_{x_t} (\|x_a - \hat{x}_a(x_t, t, \theta)\|_2^2 + \gamma \log p(x_{t-1}|x_t)), \quad (12)$$

where $\gamma$ is a hyperparameter that trades off two functions (the first one for conditional consistency and the second for better fluency). The gradient term can be interpreted as a reconstruction-based guidance, with $\eta$ controlling the strength. Following Li et al. (2022a), we repeat multiple steps of the gradient update for each diffusion step to improve the control quality. Then by replacing $\tilde{x}_a(x_t, t, \theta) := \sqrt{\bar{\alpha}_t}x_a + \sqrt{1 - \bar{\alpha}_t}\epsilon$, the sample $x_{t-1}$ will be generated using the new $\tilde{x}_0$. Algorithms in Appendix F lay out how such a sampling scheme is used for conditional generation.

## 4 EMPIRICAL EVALUATION

In this section, we first study the interpretable outputs of the proposed model. Then we evaluate our method in two modes: unconditional and conditional generation, to verify the quality of the generated signals. For time series generation, we select four previous models to compare with: TimeVAE (Desai et al., 2021), Diffwave (Kong et al., 2021), TimeGAN (Yoon et al., 2019), Cot-GAN (Xu et al., 2020) and DiffTime (Coletta et al., 2023) which can be viewed as an unconditional CSDI (Tashiro et al., 2021). We also compare to the CSDI and SSSD (Alcaraz & Strodthoff, 2022) on conditional tasks. Finally, we conduct experiments to validate the performance of Diffusion-TS when clean data is not sufficient. Implementation details and ablation study can be found in Appendix G and C.7, respectively.

## 4.1 DATASETS

We use 4 real-world datasets and 2 simulated datasets in Table 11 to evaluate our method. **Stocks** is the Google stock price data from 2004 to 2019. Each observation represents one day and has 6 features. **ETTh** dataset contains the data collected from electricity transformers, including load and oil temperature that are recorded every 15 minutes between July 2016 and July 2018. **Energy** is a UCI appliance energy prediction dataset with 28 values. **fMRI** is a benchmark for causal discovery, which consists of realistic simulations of blood-oxygen-level-dependent (BOLD) time series. Here, we select a simulation from the original dataset which has 50 features. **Sines** has 5 features where each feature is created with different frequencies and phases independently. **MuJoCo** is multivariate physics simulation time series data with 14 features.

## 4.2 METRICS

For quantitative evaluation of synthesized data, we consider three main criteria (1) the distribution similarity of time series; (2) the temporal and feature dependency; (3) the usefulness for the predictive purposes. We adopt the following evaluation metrics (see Appendix G.3 for the detailed descriptions): 1) **Discriminative score** (Yoon et al., 2019) measures the similarity using a classification model to distinguish between the original and synthetic data as a supervised task; 2) **Predictive score** (Yoon et al., 2019) measures the usefulness of the synthesized data by training a post-hoc sequence model to predict next-step temporal vectors using the train-synthesis-and-test-real (TSTR) method; 3) **Context-Fréchet Inception Distance (Context-FID) score** (Paul et al., 2022) quantifies the quality of the synthetic time series samples by computing the difference between representations of time series that fit into the local context; 4) **Correlational score** (Ni et al., 2020) uses the absolute error between cross correlation matrices by real data and synthetic data to assess the temporal dependency.

## 4.3 INTERPRETABILITY RESULTS

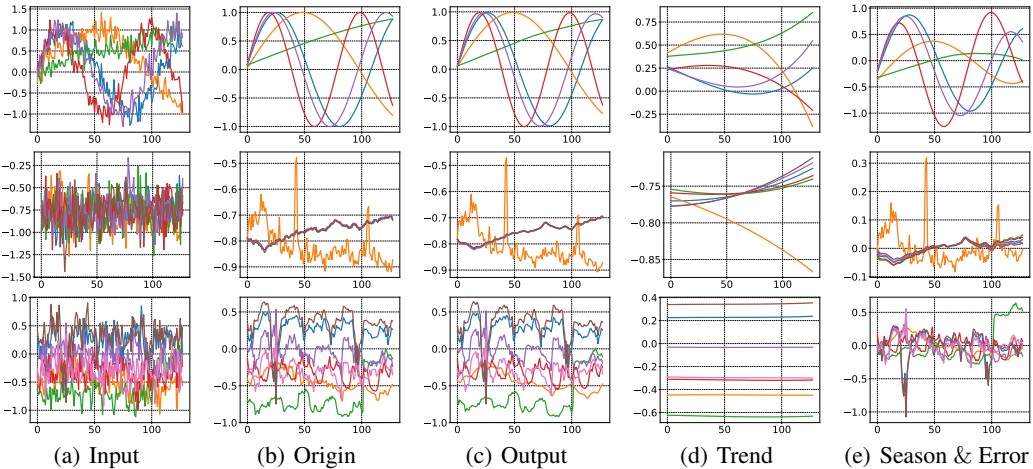

(a) Input     (b) Origin     (c) Output     (d) Trend     (e) Season & Error

Figure 3: The reconstruction performance of Diffusion-TS with the interpretable configurations. Each row is a time series example per dataset, top to bottom (Sines, Stocks and ETTh).

We start by analyzing the reconstruction performance of our model for multivariate time series. Figure 3 illustrates demos of the original and reconstructed samples by Diffusion-TS in three datasets. The model takes the corrupted samples (shown in (a)) with 50 steps of noise added as input, and outputs the signals (shown in (c)) that try to restore the ground truth (shown in (b)), with the aid of the decomposition of temporal trend (shown in (d)) and season & error (shown in (e)). As would be expected, the trend curve follows the overall shape of the signal, while the season & error oscillates around zero. Fusing the two temporal properties, the reconstructed samples can be seen a great agreement with the ground truth. Our conclusion here is that by introducing interpretable architecture, the time series generated by the Diffusion-TS can exhibit explainability of disentanglement with almost no accuracy loss. Additionally, we also conduct case study for further validating interpretability on synthetic data in Appendix C.5.

## 4.4 UNCONDITIONAL TIME SERIES GENERATION

In this part, we first replicate the experimental setup in Yoon et al. (2019) to report the performance of the model with respect to the aforementioned benchmarks. Afterwards, we investigate the ability of the models to synthesize longer time series data.

Table 1: Results on Multiple Time-Series Datasets (Bold indicates best performance).

| Metric | Methods | Sines | Stocks | ETTh | MuJoCo | Energy | fMRI |
|---|---|---|---|---|---|---|---|
| Context-FID Score 

 (Lower the Better) | Diffusion-TS | **0.006±.000** | 0.147±.025 | **0.116±.010** | **0.013±.001** | **0.089±.024** | **0.105±.006** |
| | TimeGAN | 0.101±.014 | **0.103±.013** | 0.300±.013 | 0.563±.052 | 0.767±.103 | 1.292±.218 |
| | TimeVAE | 0.307±.060 | 0.215±.035 | 0.805±.186 | 0.251±.015 | 1.631±.142 | 14.449±.969 |
| | Diffwave | 0.014±.002 | 0.232±.032 | 0.873±.061 | 0.393±.041 | 1.031±.131 | 0.244±.018 |
| | DiffTime | 0.006±.001 | 0.236±.074 | 0.299±.044 | 0.188±.028 | 0.279±.045 | 0.340±.015 |
| | Cot-GAN | 1.337±.068 | 0.408±.086 | 0.980±.071 | 1.094±.079 | 1.039±.028 | 7.813±.550 |
| Correlational Score 

 (Lower the Better) | Diffusion-TS | **0.015±.004** | **0.004±.001** | **0.049±.008** | **0.193±.027** | **0.856±.147** | **1.411±.042** |
| | TimeGAN | 0.045±.010 | 0.063±.005 | 0.210±.006 | 0.886±.039 | 4.010±.104 | 23.502±.039 |
| | TimeVAE | 0.131±.010 | 0.095±.008 | 0.111±020 | 0.388±.041 | 1.688±.226 | 17.296±.526 |
| | Diffwave | 0.022±.005 | 0.030±.020 | 0.175±.006 | 0.579±.018 | 5.001±.154 | 3.927±.049 |
| | DiffTime | 0.017±.004 | 0.006±.002 | 0.067±.005 | 0.218±.031 | 1.158±.095 | 1.501±.048 |
| | Cot-GAN | 0.049±.010 | 0.087±.004 | 0.249±.009 | 1.042±.007 | 3.164±.061 | 26.824±.449 |
| Discriminative Score 

 (Lower the Better) | Diffusion-TS | **0.006±.007** | **0.067±.015** | **0.061±.009** | **0.008±.002** | **0.122±.003** | **0.167±.023** |
| | TimeGAN | 0.011±.008 | 0.102±.021 | 0.114±.055 | 0.238±.068 | 0.236±.012 | 0.484±.042 |
| | TimeVAE | 0.041±.044 | 0.145±.120 | 0.209±.058 | 0.230±.102 | 0.499±.000 | 0.476±.044 |
| | Diffwave | 0.017±.008 | 0.232±.061 | 0.190±.008 | 0.203±.096 | 0.493±.004 | 0.402±.029 |
| | DiffTime | 0.013±.006 | 0.097±.016 | 0.100±.007 | 0.154±.045 | 0.445±.004 | 0.245±.051 |
| | Cot-GAN | 0.254±.137 | 0.230±.016 | 0.325±.099 | 0.426±.022 | 0.498±.002 | 0.492±.018 |
| Predictive Score 

 (Lower the Better) | Diffusion-TS | **0.093±.000** | **0.036±.000** | **0.119±.002** | **0.007±.000** | **0.250±.000** | **0.099±.000** |
| | TimeGAN | 0.093±.019 | 0.038±.001 | 0.124±.001 | 0.025±.003 | 0.273±.004 | 0.126±.002 |
| | TimeVAE | 0.093±.000 | 0.039±.000 | 0.126±.004 | 0.012±.002 | 0.292±.000 | 0.113±.003 |
| | Diffwave | 0.093±.000 | 0.047±.000 | 0.130±.001 | 0.013±.000 | 0.251±.000 | 0.101±.000 |
| | DiffTime | 0.093±.000 | 0.038±.001 | 0.121±.004 | 0.010±.001 | 0.252±.000 | 0.100±.000 |
| | Cot-GAN | 0.100±.000 | 0.047±.001 | 0.129±.000 | 0.068±.009 | 0.259±.000 | 0.185±.003 |
| | Original | 0.094±.001 | 0.036±.001 | 0.121±.005 | 0.007±.001 | 0.250±.003 | 0.090±.001 |

In Table 1, we list the results of the 24-length time series generation used in most of the existing related works. It shows that Diffusion-TS consistently produces higher-quality synthetic samples over other baselines in terms of almost all metrics. For example, Diffuision-TS can remarkably improve the discriminative score by an average of **50**% in all six datasets. It is also notable that for high-dimensional datasets (i.e., MuJoCo, Energy and fMRI), the leading performance of Diffusion-TS is even more significant. That demonstrates that Diffusion-TS can well tackle the challenges of complex time series synthesis. Table 3 shows the results of long-term time series generation on ETTh and Energy datasets. We randomly generate 3000 sequences with various lengths (64, 128, 256), then use aforementioned metrics to assess the generation quality of different methods. From the results, Diffusion-TS achieves the best overall performance, implying the efficacy of interpretable decomposition for long-term time-series modelling. Notably, different from other baselines, the performance of Diffusion-TS changes quite steadily as the length of sequence increases. It means Diffusion-TS retains better long-term robustness, which is meaningful for real-world applications.

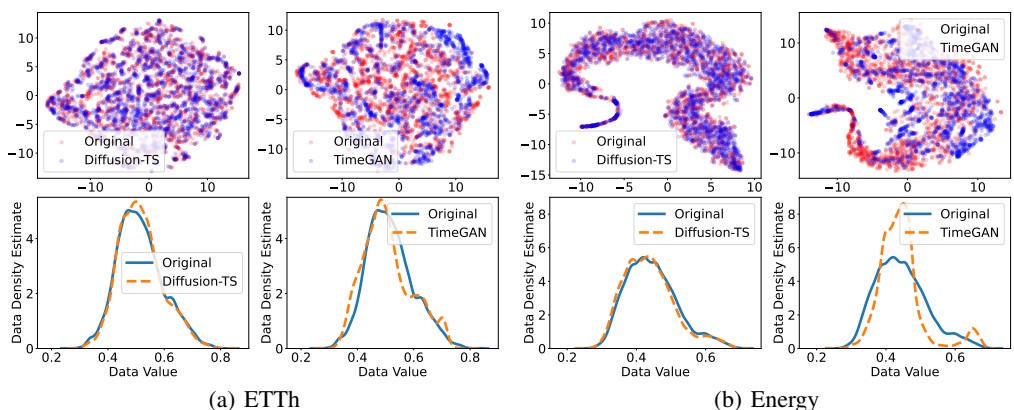

(a) ETTh  (b) Energy

Figure 4: Visualizations of the time series synthesized by Diffusion-TS and TimeGAN.

To visualize the performance of time series synthesis, we adopt two visualization methods. One is to project original and synthetic data in a 2-dimensional space using t-SNE (Van der Maaten & Hinton, 2008). The other is to draw data distributions using kernel density estimation. As shown in the $1^{st}$ row in Figure 4 and pictures in Appendix C.2, Diffusion-TS overlaps original data areas markedly better than TimeGAN. The $2^{nd}$ row in Figure 4 shows that the synthetic data's distributions from Diffusion-TS are more similar to those of the original data than TimeGAN. For more visualizations and distributions, please refer to Appendix H.

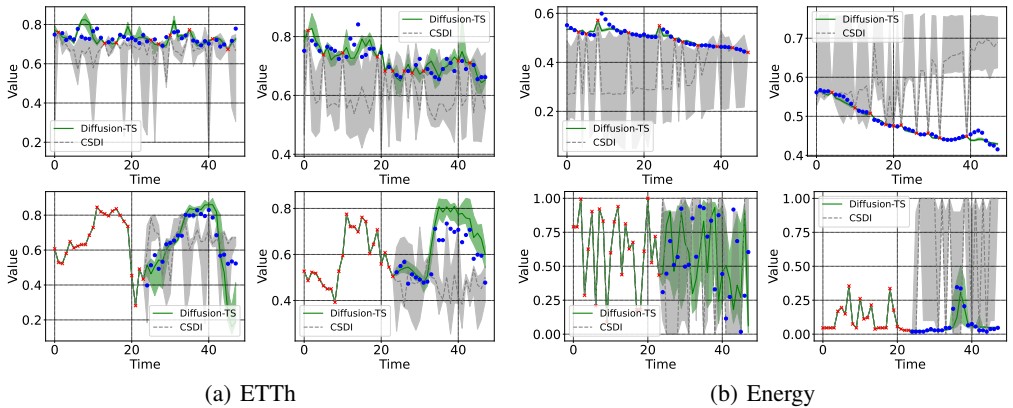

|     |     |
| --- | --- |
| (a) ETTh | (b) Energy |

Figure 5: Examples of time series imputation ($1^{st}$ row) and forecasting ($2^{nd}$ row) for ETTh and Energy datasets. Green and gray colors correspond to Diffusion-TS and CSDI, respectively.

## 4.5 CONDITIONAL TIME SERIES GENERATION

Here, we present the conditional generation on time series imputation and forecasting. For imputation tasks, we apply a masking strategy following a geometric distribution used in Zerveas et al. (2021), which controls both the lengths of the missing sequences and the missing ratio $r$, instead of selecting the missing points randomly, since a single missing time point may often be easily predicted from the immediately preceding and succeeding points. For both tasks, the length of a time series is set as 48 time steps, for which given the first $w$ continuous time points, we forecast on the remaining $48 - w$ time points. We provide imputation and forecasting examples on ETTh and Energy datasets in Figure 5 (more examples in Appendix I). The red crosses show the observed values and the blue circles show the ground-truth imputation targets. The median values of imputations are shown as the line and $5\%$ and $95\%$ quantiles are shown as the shade. It indicates that Diffusion-TS gives more reasonable imputations and predictions with high confidence against CSDI.

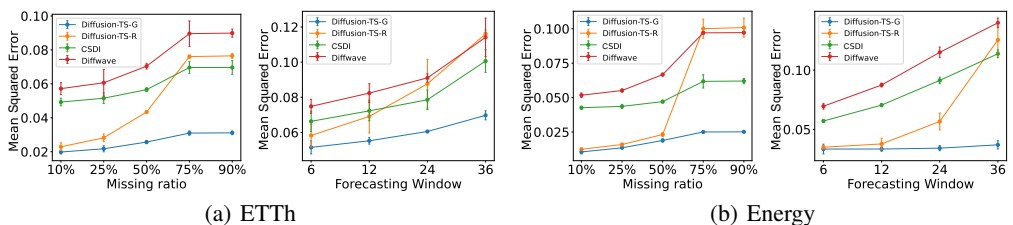

|     |     |
| --- | --- |
| (a) ETTh | (b) Energy |

Figure 6: Performance of diffusion methods for time-series imputation and forecasting. Mean and confidence intervals are obtained by running each method 5 times.

We also run detailed experiments to evaluate our conditional extension. Diffusion-TS (Diffusion-TS-G) based on the aforementioned Langevin sampler is compared with CSDI, the original Diffwave and Diffusion-TS-R using replace-based conditional sampling (see Appendix D for detailed descriptions for the conditional adaptation). In Figure 6 we show the empirical results, where Diffusion-TS-G outperformed all baselines even at high levels of missing ratio. We can observe that Diffusion-TS-R achieves close accuracy on the low missingness ratio, but for the high missing ratio $75\%$ and $90\%$, there is a clear improvement with the reconstruction-guided sampling strategy. It means that seasonal-trend decomposition can naturally facilitate infilling as restrictions increase, however,

when the missing ratio is high and there are no additional constraints, it may also make the entire sequence deviate from the ground truth. For completeness, we conduct additional experiments and include other time series baselines such as TLAE (Nguyen & Quanz, 2021) in Appendix C.3. We can see our model still performs well, as it has the best performance against the baselines.

**Cold Starts and Irregular Settings.** In this experiment, we explore whether the Diffusion-TS can keep performance from degrading under regular data deficit condition. Cold start refers to training model when little or no data is available. We use only 10/25/50/75% of the time series data in each dataset as training data respectively. Then we remove time series values from the rest dataset, leading to 10% ~ 20% missing values for a data point. After cold starts on the regular part of the dataset, we leverage the model to impute these irregular time series then continue training by incorporating them into the training set. We call this process **irregular training** and Figure 11 shows the results. We observed that even at the 10% training threshold, Diffusion-TS still has superior results on the multiple time-series datasets compared to baselines in Table 1. In addition, Diffusion-TS shows better discriminative and predictive scores in all cases after restoration. The performance of the model is always at the same level as if there were no missing data, which shows the efficacy of the Diffusion-TS under insufficient and irregular settings.

## 4.6 ABLATION STUDY

To evaluate the effectiveness of the Diffusion-TS, we compare the full versioned Diffusion-TS with its three variants: (1) w/o FFT, i.e. Diffusion-TS without Fourier-based loss term during training, (2) w/o Interpretability, i.e. Diffusion-TS without seasonal-trend design in the network, (3) w/o Transformer, i.e. Diffusion-TS based on a Convolutional network without encoder and self-attention mechanism. (4) $\epsilon$-prediction, Diffusion-TS using traditional noise prediction parameterization for training and sampling. Results are presented in Table 2 (see detailed evaluation in Appendix C.1).

Table 2: Ablation study for model architecture and options. (Bold indicates best performance).

| Metric | Method | Sines | Stocks | ETTh | Mujoco | Energy | fMRI |
|---|---|---|---|---|---|---|---|
| Discriminative Score (Lower the Better) | Diffusion-TS | **0.006±.007** | **0.067±.015** | **0.061±.009** | **0.008±.002** | **0.122±.003** | 0.167±.023 |
| | w/o FFT | 0.007±.006 | 0.127±.019 | 0.096±.007 | 0.010±.002 | 0.135±.004 | 0.177±.013 |
| | w/o Interpretability | 0.009±.006 | 0.101±.096 | 0.071±.010 | 0.021±.014 | 0.125±.003 | 0.267±.034 |
| | w/o Transformer | 0.010±.007 | 0.104±.024 | 0.082±.006 | 0.039±.014 | 0.324±.015 | **0.123±.064** |
| | $\epsilon$-prediction | 0.040±.011 | 0.131±.014 | 0.099±.010 | 0.023±.006 | 0.197±.001 | 0.168±.030 |
| Predictive Score (Lower the Better) | Diffusion-TS | **0.093±.000** | **0.036±.000** | **0.119±.002** | **0.007±.000** | **0.250±.000** | **0.099±.000** |
| | w/o FFT | **0.093±.000** | 0.038±.000 | 0.121±.004 | 0.008±.001 | **0.250±.000** | 0.100±.001 |
| | w/o Interpretability | **0.093±.000** | 0.037±.000 | **0.119±.008** | 0.008±.001 | 0.251±.000 | 0.101±.000 |
| | w/o Transformer | **0.093±.000** | **0.036±.000** | 0.126±.004 | 0.008±.000 | 0.319±.006 | **0.099±.000** |
| | $\epsilon$-prediction | 0.097±.000 | 0.039±.000 | 0.120±.002 | 0.008±.001 | 0.251±.000 | 0.100±.000 |
| | Original | 0.094±.001 | 0.036±.001 | 0.121±.005 | 0.007±.001 | 0.250±.003 | 0.090±.001 |

We can find that the best result on each dataset is usually obtained with Diffusion-TS. Removing the attention and residuals often causes big performance drops. However, when the dataset, i.e. fMRI, has a high frequency and dimension, a network with only an interpretable design achieves the most significant performance improvement. In addition, the FFT-loss term and signal prediction parameterization show a crucial role in general. The conclusion is that our design is relatively steady across all experiment settings. And the decomposition not only adds interpretable capability without losing accuracy, but also does boost the unconditional generation of the diffusion model when there are obvious frequency fluctuations in the data set.

## 5 CONCLUSIONS

In this paper, we have proposed Diffusion-TS, a DDPM-based method for general time series generation. In addition to the generative capability of DDPMs, the Diffusion-TS is powered by the TS-specific loss design and transformer-based deep decomposition architecture. Besides, our model trained for unconditional generation could be readily extended for conditional generation by combining gradients into the sampling. The experiments have shown our model is capable of a wide range of time-series generative tasks and can achieve competitive performance. Shown in Table 8, one notable limitation of DDPMs is the high cost of inference, which is demanding for more computing resources to generate a sample compared with GAN-based approaches, although our underlying model is lightweight anyway. To improve Diffusion-TS for which both conditional and unconditional inference procedures converge in shorter time would be a potential future work.

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

## A   RELATED WORK

**Time Series Generation.** Deep generative models of all kinds have exhibited high-quality samples in a wide variety of domains, where time series generation is one of the most challenging tasks in the generative research field. Existing literature for time-series synthesis is based predominantly on generative adversarial networks (GANs), and many GAN-based architectures use recurrent networks to model temporal dynamics (Mogren, 2016; Esteban et al., 2017; Yoon et al., 2019; Pei et al., 2021; Jeon et al., 2022). Among these GAN-based works, Mogren (2016) first introduced a GAN structure with LSTM called C-RNN-GAN. Then a Recurrent Conditional GAN (RCGAN) (Esteban et al., 2017) was proposed which uses a basic RNN as generator and discriminator and auxiliary label information as a condition to generate medical time series. Yoon et al. developed TimeGAN (Yoon et al., 2019) by adding an embedding function and supervised loss to the original GAN for capturing the temporal dynamics of data throughout time. In addition, Pei et al. (2021) proposed RTSGAN, and Paul et al. (2022) designed PSA-GAN that generates long univariate time series samples of high quality using progressive growing of GANs along with self-attention.

Due to the instabilities typical of adversarial objectives, another line of work in the time series field recently focuses on other deep generative methods. Jarrett et al. (2021) imitates the sequential behavior of time series using a stepwise-decomposable energy model as reinforcement. Fourier Flows (Alaa et al., 2021) was proposed as a method based on normalizing flows followed by a chain of spectral filters leading to an exact likelihood optimization. Desai et al. (2021) introduced TimeVAE which implements an interpretable temporal structure, and achieves reasonable results on time series synthesis with Variational Autoencoder (VAE). Most recently, Implicit neural representations (INRs) have also been used for time series generation. Fons et al. (2022) leverages the latent embeddings learned by the hypernetwork for the synthesis of new time series by interpolation.

**Interpretable Time Series Decomposition.** Seasonal-trend decomposition is an effective method used to decompose time series into seasonal, trend components and remainder components to attain interpretability in time series analysis (Cleveland et al., 1990; De Livera et al., 2011; Scott & Varian, 2015; Jammalamadaka et al., 2018; Dokumentov & Hyndman, 2021; Theodosiou, 2011; Yang et al., 2015; Wen et al., 2018; 2020). The classic approach for performing the decomposition is the widely used STL algorithm (Cleveland et al., 1990). To handle multiple seasonality and seasonal shifts, STR (Dokumentov & Hyndman, 2021) has been proposed by jointly extracting trend, seasonal, and remainder components without iterations. De Livera et al. (2011) introduced a framework based on the state space model for complex seasonal time series. More recently, a novel seasonal-trend decomposition called RobustSTL and its extended work (Wen et al., 2018; 2020) have been proposed. The ability to break time series into interpretable components has been a topic of recent interest in the field of anomaly detection (Zancato et al., 2022), forecasting (Zhou et al., 2022; Wu et al., 2021), and generation (Desai et al., 2021). This work follows N-BEATS (Oreshkin et al., 2020), a univariate time series forecasting architecture that explicitly encodes seasonal-trend decomposition into the network by defining two blocks respectively.

**Denoising Diffusion Probabilistic Models.** Denoising diffusion probabilistic models (DDPMs) are a new class of generative models with nice properties (Ho et al., 2020). They have demonstrated great success in both continuous and discrete data domains, producing images (Ho et al., 2020; Dhariwal & Nichol, 2021; Gu et al., 2022), text (Li et al., 2022a; Yu et al., 2022; Strudel et al., 2022), audio (Kong et al., 2021), and video (Luo et al., 2023; Harvey et al., 2022; Ho et al., 2022) that have state-of-the-art sample quality. Very recently, diffusion models have also been developed for time series data. TimeGrad (Rasul et al., 2021) is a conditional diffusion model which predicts in an autoregressive manner, with the denoising process guided by the hidden state of a recurrent neural network. CSDI (Tashiro et al., 2021) and SSSD (Alcaraz & Strodthoff, 2022) use a self-supervised masking to guide the denoising process like image inpainting. To alleviate the problem of boundary disharmony, the non-autoregressive diffusion model TimeDiff (Shen & Kwok, 2023) uses future mixup and autoregressive initialization for conditioning. Regarding unconditional generation, Lim et al. (2023) employs recurrent neural networks as the backbone to product regular 24-time-series using SGM. And most recently, based on the structured state space model used in SSSD, Kollovieh et al. (2023) conducts univariate time series generation and forecasting with self-guiding strategy. Meanwhile, Coletta et al. (2023) approximated the diffusion function based on CSDI where

they remove the side information provided as embedding. They also introduced GuidedDiffTime to handle new constraints such as trend or fixed-value, without re-training.

# B  DENOISING DIFFUSION PROBABILISTIC MODELS (DDPMs)

In this section, we provide a brief overview of DDPMs. On a high level, diffusion models sample from a distribution by reversing a gradual noising process. In particular, the forward process $q$ gradually corrupts original data $x_0 \in \mathbb{R}^d$ via a fixed Markov chain $x_0, \ldots, x_T$ with each variable in $\mathbb{R}^d$ as follows:

$$q(x_t|x_{t-1}) := \mathcal{N}(x_t; \sqrt{1 - \beta_t}x_{t-1}, \beta_t I), \quad q(x_{1:T}|x_0) := \prod_{t=1}^{T} q(x_t|x_{t-1}), \tag{13}$$

where $\beta_t \in (0, 1)$ is a variance at diffusion step $t$. The increasingly noisy variables $x_{1...T}$ have the same dimensionality as $x_0$, and $x_T$ is an isotropic Gaussian.

A notable property of the forward process is that using notation $\alpha_t := 1 - \beta_t$ and $\bar{\alpha}_t := \prod_{s=1}^{t} \alpha_s$, we can sample $x_t$ at any arbitrary time step $t$ in a closed form:

$$q(x_t|x_0) := \mathcal{N}(x_t; \sqrt{\bar{\alpha}_t}x_0, (1 - \bar{\alpha}_t)I). \tag{14}$$

Thus using reparameterization trick Kingma & Welling (2013) and defining $\epsilon \sim \mathcal{N}(0, I)$, we have

$$x_t = \sqrt{\bar{\alpha}_t}x_0 + \sqrt{1 - \bar{\alpha}_t}\epsilon. \tag{15}$$

Starting from the noise $x_T$, we can run the reverse process parametrized by the model $p_\theta(x_{t-1}|x_t) := \mathcal{N}(x_{t-1}; \mu_\theta(x_t, t), \Sigma_\theta(x_t, t))$ to get $x_0$. The diffusion model is trained to maximize the marginal likelihood of the data $\mathbb{E}_{x_0}[\log p_\theta(x_0)]$, and we can write the variational lower bound (VLB) as follows:

$$\mathcal{L}_{vlb} := \underbrace{-\log p_\theta(x_0|x_1)}_{\mathcal{L}_0} + \sum_{t=2}^{T} \underbrace{D_{KL}(q(x_{t-1}|x_t, x_0) \,||\, p_\theta(x_{t-1}|x_t))}_{\mathcal{L}_{t-1}} + \underbrace{D_{KL}(q(x_T|x_0) \,||\, p(x_T))}_{\mathcal{L}_T}. \tag{16}$$

The main objective is a sum of independent terms $\mathcal{L}_{t-1}$. There are many different ways to parameterize $\mu_\theta(x_t, t)$, and the most obvious option is to predict $\mu_\theta(x_t, t)$ directly:

$$\mathcal{L}_{origin} := \mathbb{E}_{t,x_0} \left[ \frac{1}{2\Sigma_\theta^2(x_t, t)} \|\hat{\mu}(x_t, x_0) - \mu_\theta(x_t, t)\|^2 \right], \tag{17}$$

where $\hat{\mu}(x_t, x_0)$ is the mean of the posterior $q(x_t|x_{t-1}, x_0)$ which are defined as follows:

$$q(x_{t-1}|x_t, x_0) = \mathcal{N}(x_{t-1}; \frac{\sqrt{\bar{\alpha}_{t-1}}\beta_t}{1 - \bar{\alpha}_t}x_0 + \frac{\sqrt{\alpha_t}(1 - \bar{\alpha}_{t-1})}{1 - \bar{\alpha}_t}x_t, \frac{1 - \bar{\alpha}_{t-1}}{1 - \bar{\alpha}_t}\beta_t \mathbf{I}). \tag{18}$$

Finally, Ho et al. (2020) found predicting $\epsilon$ worked best, especially when combined with a reweighted loss function:

$$\mathcal{L}_{simple} = \mathbb{E}_{t,x_0,\epsilon} \left[ \|\epsilon - \epsilon_\theta(x_t, t)\|^2 \right]. \tag{19}$$

But they also admitted that there is the possibility of predicting $x_0$ directly. Note that estimating $\epsilon$ in Equation 19 is equivalent to estimating a scaled version of the score function, i.e. the gradient of the log density of the noisy data:

$$\nabla_{x_t} \log q_t(x_t|x_0) \approx -\frac{1}{1 - \bar{\alpha}_t}(x_t - \sqrt{\bar{\alpha}_t}x_0) = -\frac{1}{\sqrt{1 - \bar{\alpha}_t}}\epsilon. \tag{20}$$

Thus, data generation through denoising depends on the score-function and can be seen as noise conditional score-based generation Voleti et al. (2022). Then the reverse process $p_\theta(x_{t-1}|x_t)$ in DDPM can be written as:

$$x_{t-1} = \frac{1}{\sqrt{\alpha_t}}(x_t - \frac{1 - \alpha_t}{\sqrt{1 - \bar{\alpha}_t}}\epsilon_\theta(x_t, t)) + \sigma_t z, \tag{21}$$

where $\sigma_t$ is hyperparameter, and $z$ is a standard Gaussian noise.

## C ADDTIONAL EXPERIMENTAL RESULTS

In this section we present additional experiments which we omitted in the main body of the paper due to limited space.

### C.1 DETAILED EXPERIMENTS FOR LONG-TERM TIME SERIES GENERATION

In order to further verify its performance stability in producing long multivariate time-series data, we add discriminative score and predictive score:

Table 3: Detailed Results of Long-term Time-series Generation. (Bold indicates best performance).

| | Dataset | Length | Diffusion-TS | TimeGAN | TimeVAE | Diffwave | DiffTime | Cot-GAN |
|---|---|---|---|---|---|---|---|---|
| ETTh | Context-FID Score (Lower the Better) | 64 | **0.631±.058** | 1.130±.102 | 0.827±.146 | 1.543±.153 | 1.279±.083 | 3.008±.277 |
| | | 128 | **0.787±.062** | 1.553±.169 | 1.062±.134 | 2.354±.170 | 2.554±.318 | 2.639±.427 |
| | | 256 | **0.423±.038** | 5.872±.208 | 0.826±.093 | 2.899±.289 | 3.524±.830 | 4.075±.894 |
| | Correlational Score (Lower the Better) | 64 | 0.082±.005 | 0.483±.019 | **0.067±.006** | 0.186±.008 | 0.094±.010 | 0.271±.007 |
| | | 128 | 0.088±.005 | 0.188±.006 | **0.054±.007** | 0.203±.006 | 0.113±.012 | 0.176±.006 |
| | | 256 | 0.064±.007 | 0.522±.013 | **0.046±.007** | 0.199±.003 | 0.135±.006 | 0.222±.010 |
| | Discriminative Score (Lower the Better) | 64 | **0.106±.048** | 0.227±.078 | 0.171±.142 | 0.254±.074 | 0.150±.003 | 0.296±.348 |
| | | 128 | **0.144±.060** | 0.188±.074 | 0.154±.087 | 0.274±.047 | 0.176±.015 | 0.451±.080 |
| | | 256 | **0.060±.030** | 0.442±.056 | 0.178±.076 | 0.304±.068 | 0.243±.005 | 0.461±.010 |
| | Predictive Score (Lower the Better) | 64 | **0.116±.000** | 0.132±.008 | 0.118±.004 | 0.133±.008 | 0.118±.004 | 0.135±.003 |
| | | 128 | **0.110±.003** | 0.153±.014 | 0.113±.005 | 0.129±.003 | 0.120±.008 | 0.126±.001 |
| | | 256 | **0.109±.013** | 0.220±.008 | 0.110±.027 | 0.132±.001 | 0.118±.003 | 0.129±.000 |
| Energy | Context-FID Score (Lower the Better) | 64 | **0.135±.017** | 1.230±.070 | 2.662±.087 | 2.697±.418 | 0.762±.157 | 1.824±.144 |
| | | 128 | **0.087±.019** | 2.535±.372 | 3.125±.106 | 5.552±.528 | 1.344±.131 | 1.822±.271 |
| | | 256 | **0.126±.024** | 5.032±.831 | 3.768±.998 | 5.572±.584 | 4.735±.729 | 2.533±.467 |
| | Correlational Score (Lower the Better) | 64 | **0.672±.035** | 3.668±.106 | 1.653±.208 | 6.847±.083 | 1.281±.218 | 3.319±.062 |
| | | 128 | **0.451±.079** | 4.790±.116 | 1.820±.329 | 6.663±.112 | 1.376±.201 | 3.713±.055 |
| | | 256 | **0.361±.092** | 4.487±.214 | 1.279±.114 | 5.690±.102 | 1.800±.138 | 3.739±.089 |
| | Discriminative Score (Lower the Better) | 64 | **0.078±.021** | 0.498±.001 | 0.499±.000 | 0.497±.004 | 0.328±.031 | 0.499±.001 |
| | | 128 | **0.143±.075** | 0.499±.001 | 0.499±.000 | 0.499±.001 | 0.396±.024 | 0.499±.001 |
| | | 256 | **0.290±.123** | 0.499±.000 | 0.499±.000 | 0.499±.000 | 0.437±.095 | 0.498±.004 |
| | Predictive Score (Lower the Better) | 64 | **0.249±.000** | 0.291±.003 | 0.302±.001 | 0.252±.001 | 0.252±.000 | 0.262±.002 |
| | | 128 | **0.247±.001** | 0.303±.002 | 0.318±.000 | 0.252±.000 | 0.251.±.000 | 0.269±.002 |
| | | 256 | **0.245±.001** | 0.351±.004 | 0.353±.003 | 0.251±.000 | 0.251±.000 | 0.275±.004 |

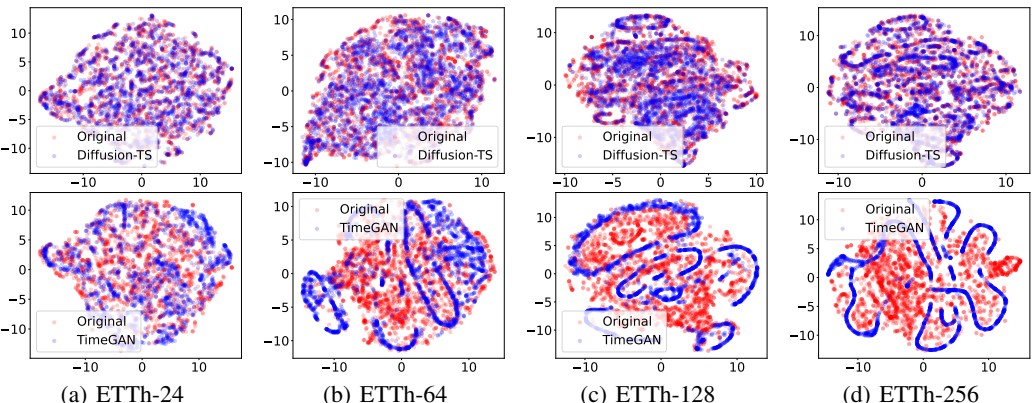

(a) ETTh-24   (b) ETTh-64   (c) ETTh-128   (d) ETTh-256

Figure 7: t-SNE plots of the time series of length 24, 64, 128 and 256 synthesized by Diffusion-TS and TimeGAN. Red dots represent real data instances, and blue dots represent generated data samples in all plots.

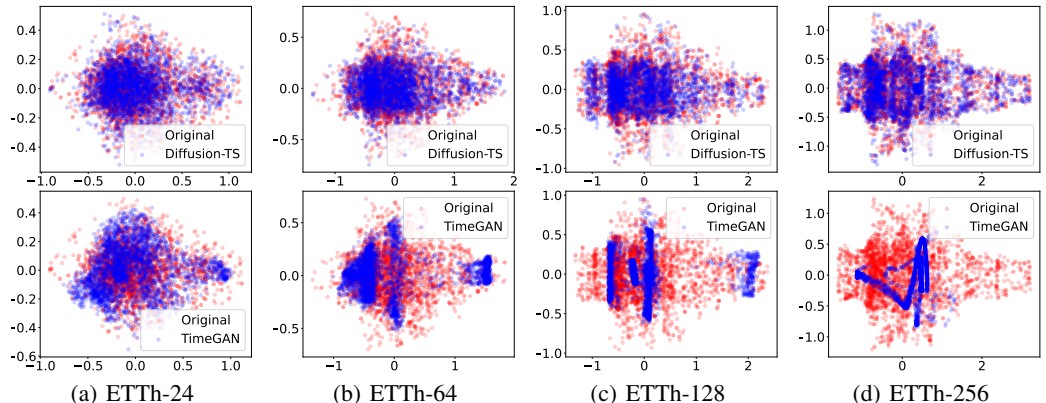

|   (a) ETTh-24   |   (b) ETTh-64   |   (c) ETTh-128   |   (d) ETTh-256   |

Figure 8: PCA plots of the time series of length 24, 64, 128 and 256 synthesized by Diffusion-TS and TimeGAN. Red dots represent real data instances, and blue dots represent generated data samples in all plots.

## C.2   ADDITIONAL 2-DIMENSIONAL PLOTS ON ETTH DATASET

In order to further compare the diversity of the generated time-series using Diffusion-TS, and TimeGAN with the ETTh dataset, we applied PCA and t-SNE analyses to visualize how well the generated time-series distributions cover the real data distributions. According to the figures, synthetic samples generated by Diffusion-TS have significantly more overlap with the original data than the SOTA method.

## C.3   ADDITIONAL EXPERIMENTS FOR IMPUTATION AND FORECASTING PERFORMANCE COMPARISON

To demonstrate that the excellent performance of Diffusion-TS extends to further baselines, we now repeat imputation and forecasting experiments in Alcaraz & Strodthoff (2022). All baselines results were collected from the original publications. We first conduct imputation experiments on the Mu-JoCo sequences of length 100. We report an averaged MSE for a single imputation per sample on the test set over 5 trials. Table 4 shows the empirical results on the MuJoCo data set, where Diffusion-TS outperforms all baselines except for CSDI on the 70% missing ratio. We then demonstrate Diffusion-TS's forecasting capabilities on the Solar data set collected from GluonTS (Alexandrov et al., 2020), where the conditional values and forecast horizon are 168 and 24 time steps respectively. The accuracy of all models on the long time series is shown in Table 5. Similar to the observation made in imputation tasks, our proposed method can achieve significantly better performance. These results further support the effectiveness of our proposed model, as well as its ability in processing long time-series.

Table 4: Imputation MSE results for the MuJoCo data set. Here, we use a concise error notation where the values in brackets affect the least significant digits e.g. 0.572(12) signifies 0.572 ± 0.012. Similarly, all MSE results are in the order of 1e-3.

| Model | 70% Missing | 80% Missing | 90% Missing |
|---|---|---|---|
| RNN GRU-D | 11.34 | 14.21 | 19.68 |
| ODE-RNN | 9.86 | 12.09 | 16.47 |
| NeuralCDE | 8.35 | 10.71 | 13.52 |
| Latent-ODE | 3 | 2.95 | 3.6 |
| NAOMI | 1.46 | 2.32 | 4.42 |
| NRTSI | 0.63 | 1.22 | 4.06 |
| CSDI | **0.24(3)** | 0.61(10) | 4.84(2) |
| SSSD | 0.59(8) | 1.00(5) | 1.90(3) |
| Diffusion-TS | 0.37(3) | **0.43(3)** | **0.73(12)** |

Table 5: Time series forecasting results for the solar data set.

| Model | MSE |
|---|---|
| GP-copula | 9.8e2±5.2e1 |
| TransMAF | 9.30E+02 |
| TLAE | 6.8e2±7.5e1 |
| CSDI | 9.0e2±6.1e1 |
| SSSD | 5.03e2±1.06e1 |
| Diffusion-TS | **3.75e2±3.6e1** |

Figure 9 and 10 shows example prediction results on solar and stock data set respectively. As can be seen, Diffusion-TS produces high-quality predictions close to the ground truth.

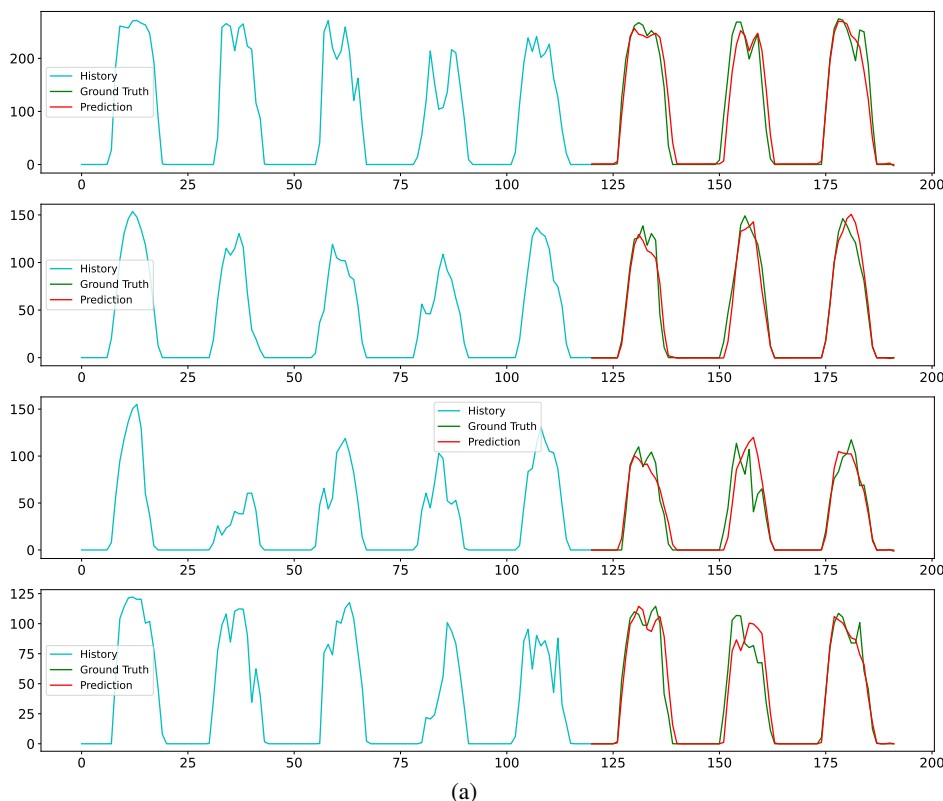

(a)

Figure 9: Visualizations of time series forecasting on Solar dataset by the proposed Diffusion-TS.

## C.4 IRREGULAR TRAINING

In this subsection, we provide the irregular training results on four real-world datasets mentioned in the main paper in Figure 11.

## C.5 DISENTANGLEMENT VALIDATION ON SYNTHETIC DATASET

In order to validate Diffusion-TS's interpretability shown in Equation 2, we conduct experiments on synthetic time-series where a time-series is composed of seasonal part and trend-cyclical part. We generate 2000 time series, each consisting of 160 time steps for training. As shown in Figure 12, we visualize the ground truth trend and seasonality patterns, together with the learned ones. We can clearly find that the learned disentangled components are very similar to the ground truth, and residual part is close to zero. This verifies the interpretability of our proposed progressive decomposition architecture.

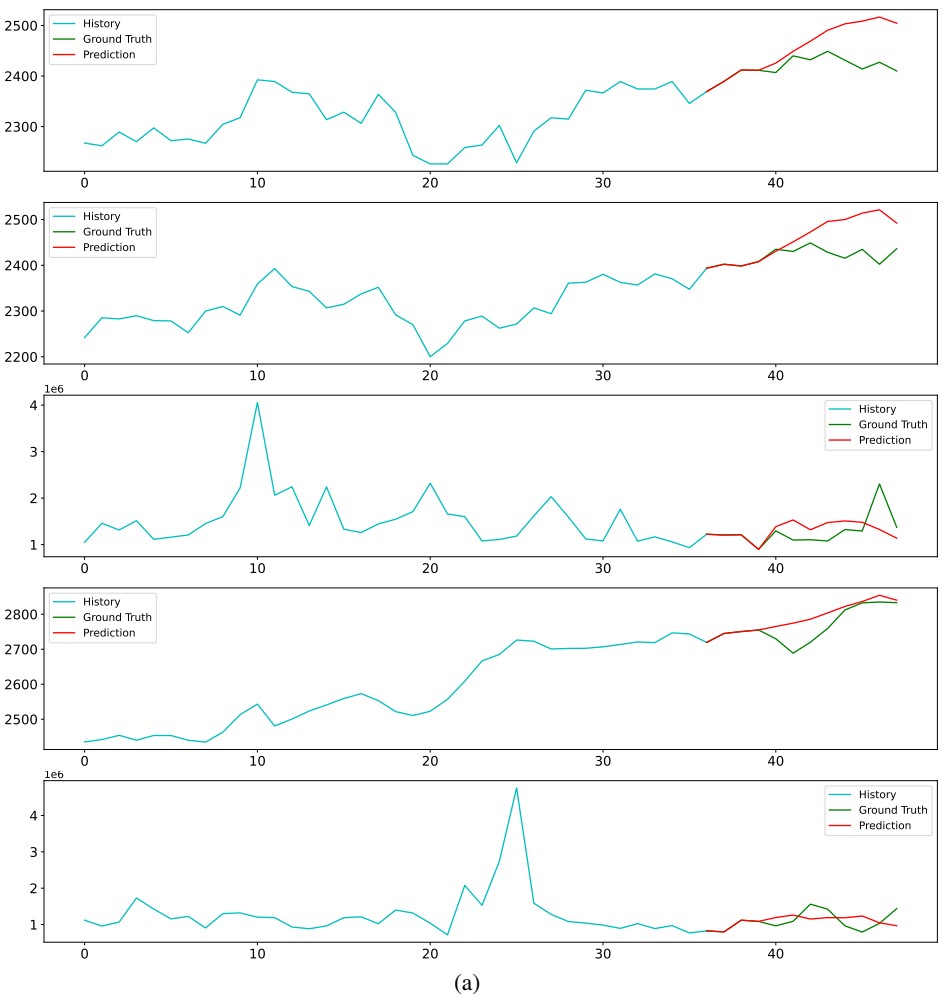

(a)

Figure 10: Visualizations of time series forecasting on Stock dataset by the proposed Diffusion-TS.

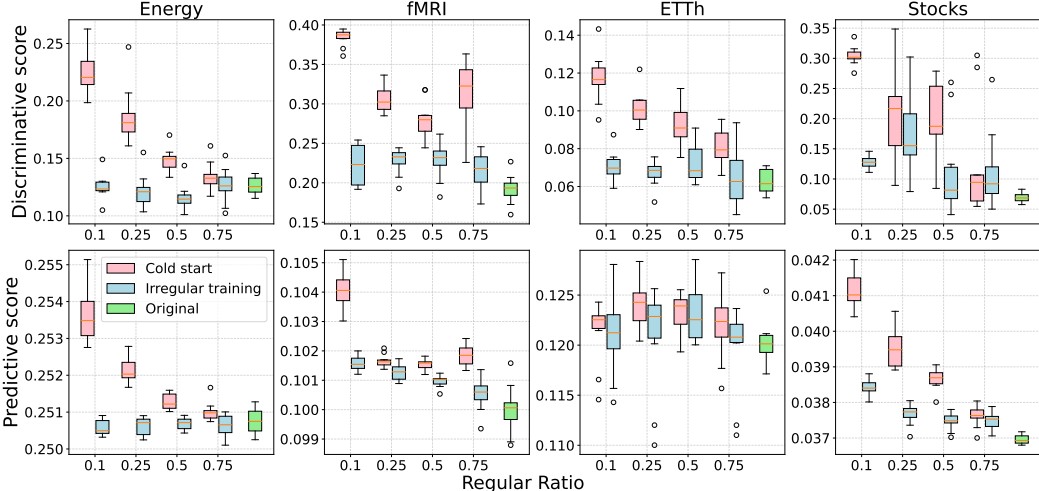

Figure 11: Performance of using different regular ratios in the cold start and irregular training experiments with $10\% \sim 20\%$ missing. Cold start only uses clean data, while the other restores and incorporates irregular time series during training. Overall, Diffusion-TS can mostly achieve the quality of no missing values via irregular training.

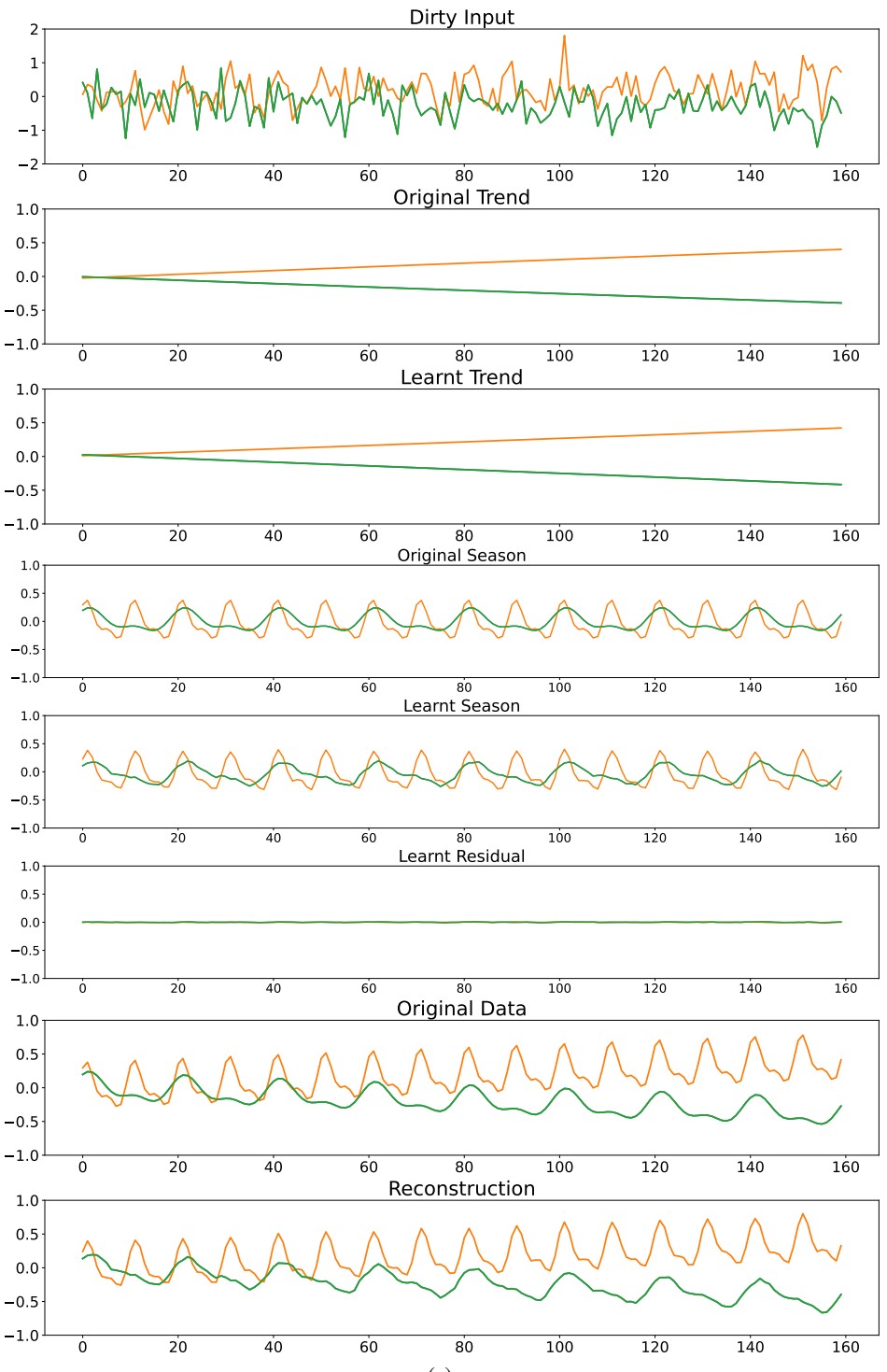

(a)

Figure 12: Disentanglement validation on synthetic dataset.

## C.6 HYPERPARAMETER TUNING AND SENSITIVITY

We did limited hyperparameter tuning in this study to find default hyperparemters that perform well across datasets. The range considered for each hyper-parameter is: batch size : [32; 64; 128], the number of attention heads: [4; 8], the number of basic dimension: [32, 64, 96, 128], the diffusion steps: [50, 200, 500, 1000] and the guidance strength: [1., 1e-1, 5e-2, 1e-2, 1e-3]. In table 6 we evaluate the impact of the different guidance strength in the model according the mean squared errors on the MuJoCo data set. Also in this table, we notice that the increasing or reducing the strength do not improve the results of conditional generation. A single Nvidia 3090 GPU is used for model training. In all of our experiments, we use $cosine$ noise scheduling and optimize our network using Adam with $(\beta_1, \beta_2) = (0.9, 0.96)$. And a linearly decay learning rate starts at 0.0008 after 500 iterations of warmup. For conditional generation, we set the inference steps, $\gamma$ to be 200, 0.05 respectively. We use 90% of the dataset for training and the rest for testing. In Table 7, we report the model size of our method and baselines. Diffusion models have generally more parameters than other methods, but our method has fewer parameters than Diffwave. And in Table 8, we list the detailed hyperparameter settings.

Table 6: Diffusion-TS with different guidance strengths $\gamma \in$ [1., 1e-1, 5e-2, 1e-2, 1e-3].

| $\gamma$ | 70% Missing | 80% Missing | 90% Missing |
|---|---|---|---|
| 1. | 2.8(13) | 4.1(10) | 6.8(17) |
| 1e-1 | **0.37(4)** | 0.45(0) | 0.82(9) |
| 5e-2 | **0.37(3)** | **0.43(3)** | **0.73(12)** |
| 1e-2 | 0.60(5) | 0.70(10) | 1.07(14) |
| 1e-3 | 3.1(8) | 7.2(20) | 19.6(22) |

Table 7: Comparison of model size.

| Model | Sines | Stocks | Energy |
|---|---|---|---|
| TimeVAE | 97,525 | 104,412 | 677,418 |
| TimeGAN | 34,026 | 48,775 | 1,043,179 |
| Cot-GAN | 40,133 | 52,675 | 601,539 |
| Diffwave | 533,592 | 599,448 | 1,337,752 |
| Diffusion-TS | 232,177 | 291,318 | 1,135,144 |

Table 8: Hyperparameters, training details, and compute resources used for each model

| Parameter | Sines | Stocks | ETTh | MuJoCo | Energy | fMRI |
|---|---|---|---|---|---|---|
| attention heads | 4 | 4 | 4 | 4 | 4 | 4 |
| attention head dimension | 16 | 16 | 16 | 16 | 24 | 24 |
| encoder layers | 1 | 2 | 3 | 3 | 4 | 4 |
| decoder layers | 2 | 2 | 2 | 2 | 3 | 4 |
| batch size | 128 | 64 | 128 | 128 | 64 | 64 |
| timesteps / sampling steps | 500 | 500 | 500 | 1000 | 1000 | 1000 |
| training steps | 12000 | 10000 | 18000 | 14000 | 25000 | 15000 |
| model size | 232,177 | 291,318 | 350,459 | 357,214 | 1,135,144 | 1,382,290 |
| training time | 17min | 15min | 31min | 25min | 60min | 44min |
| sampling time (every 2000) | 23s | 26s | 31s | 50s | 65s | 72s |

## C.7 ABLATION STUDY

To evaluate the effectiveness of the Diffusion-TS, we compare the full versioned Diffusion-TS with its three variants: (1) w/o FFT, i.e. Diffusion-TS without Fourier-based loss term during training, (2) w/o Interpretability, i.e. Diffusion-TS without seasonal-trend design in the network, (3) w/o

Transformer, i.e. Diffusion-TS based on a Convolutional network without encoder and self-attention mechanism. (4) $\epsilon$-prediction, Diffusion-TS using traditional noise prediction parameterization for training and sampling. Here shown as Table 9, we provide ablation study on 24-length time series in a detailed version.

Table 9: Ablation study for model architecture and options. (Bold indicates best performance).

| Metric | Method | Sines | Stocks | ETTh | Mujoco | Energy | fMRI |
|---|---|---|---|---|---|---|---|
| Context-FID Score | Diffusion-TS | **0.006±.000** | **0.147±.025** | **0.116±.010** | **0.013±.001** | **0.089±.024** | 0.105±.006 |
| | w/o FFT | 0.008±.001 | 0.216±.031 | 0.219±.021 | 0.017±.002 | 0.109±.014 | 0.189±.003 |
| | w/o Interpretability | 0.008±.000 | 0.181±.029 | 0.121±.004 | 0.018±.002 | 0.105±.020 | 0.149±.010 |
| | w/o Transformer | 0.007±.000 | 0.254±.024 | 0.229±.024 | 0.040±.003 | 0.496±.088 | **0.100±.010** |
| (Lower the Better) | $\epsilon$-prediction | 0.019±.004 | 0.215±.033 | 0.265±.008 | 0.017±.001 | 0.189±.054 | 0.190±.003 |
| Correlational Score | Diffusion-TS | **0.015±.004** | **0.004±.001** | **0.049±.008** | **0.193±.027** | **0.856±.147** | 1.411±.042 |
| | w/o FFT | 0.016±.005 | 0.005±.005 | 0.063±.012 | 0.204±.017 | 0.891±.023 | 1.625±.021 |
| | w/o Interpretability | 0.016±.003 | 0.010±.007 | 0.067±.009 | 0.199±.026 | 1.013±.088 | 1.340±.050 |
| | w/o Transformer | 0.016±.005 | 0.042±.007 | 0.055±.005 | 0.254±.060 | 2.164±.057 | **1.061±.041** |
| (Lower the Better) | $\epsilon$-prediction | 0.024±.006 | 0.007±.006 | 0.065±.008 | 0.208±.016 | 1.111±.114 | 1.166±.031 |
| Discriminative Score | Diffusion-TS | **0.006±.007** | **0.067±.015** | **0.061±.009** | **0.008±.002** | **0.122±.003** | 0.167±.023 |
| | w/o FFT | 0.007±.006 | 0.127±.019 | 0.096±.007 | 0.010±.002 | 0.135±.004 | 0.177±.013 |
| | w/o Interpretability | 0.009±.006 | 0.101±.096 | 0.071±.010 | 0.021±.014 | 0.125±.003 | 0.267±.034 |
| | w/o Transformer | 0.010±.007 | 0.104±.024 | 0.082±.006 | 0.039±.014 | 0.324±.015 | **0.123±.064** |
| (Lower the Better) | $\epsilon$-prediction | 0.040±.011 | 0.131±.014 | 0.099±.010 | 0.023±.006 | 0.197±.001 | 0.168±.030 |
| Predictive Score | Diffusion-TS | **0.093±.000** | **0.036±.000** | **0.119±.002** | **0.007±.000** | **0.250±.000** | **0.099±.000** |
| | w/o FFT | **0.093±.000** | 0.038±.000 | 0.121±.004 | 0.008±.001 | **0.250±.000** | 0.100±.001 |
| | w/o Interpretability | **0.093±.000** | 0.037±.000 | **0.119±.008** | 0.008±.001 | 0.251±.000 | 0.101±.000 |
| | w/o Transformer | **0.093±.000** | **0.036±.000** | 0.126±.004 | 0.008±.000 | 0.319±.006 | **0.099±.000** |
| (Lower the Better) | $\epsilon$-prediction | 0.097±.000 | 0.039±.000 | 0.120±.002 | 0.008±.001 | 0.251±.000 | 0.100±.000 |
| | Original | 0.094±.001 | 0.036±.001 | 0.121±.005 | 0.007±.001 | 0.250±.003 | 0.090±.001 |

Table 10: Ablation analysis of the decomposition on MuJoCo dataset.

| Model | 70% Missing | 80% Missing | 90% Missing |
|---|---|---|---|
| Residual | 0.51(1) | 0.59(7) | 0.85(10) |
| Residual+Season | 0.45(5) | 0.52(3) | 0.77(9) |
| Residual+Trend | 0.46(2) | 0.50(5) | 0.80(7) |
| Season+Trend | 0.63(3) | 1.05(6) | 1.42(10) |
| Diffusion-TS | **0.37(3)** | **0.43(3)** | **0.73(12)** |

We also conduct a fine-grained ablation study to verify the effectiveness of our proposed disentangled framework. The ablative results are shown in Table 10. We find that the model performance drops no matter which part is removed, which validates that all of our proposed disentangled representations play an important role in generative tasks and jointly enhance the performance of the final model.

## D   REPLACE-BASED IMPUTATION WITH DIFFUSION MODEL

We now introduce the imputation method with Diffwave and Diffusion-TS-R used for the experiments in Section 4.5. Given an irregular sample $x = [x^{\mathrm{ob}}, x^{\mathrm{ta}}]$ with $x^{\mathrm{ob}}$ denotes all observed values and $x^{\mathrm{ta}}$ denotes all missing values, Song et al. (2020) proposed a general method for conditional sampling from the jointly trained diffusion model $p_\theta(x)$. By defining the known and unknown dimensions of $x_t$ as $\Omega(x_t)$ and $\bar{\Omega}(x_t)$ respectively, they have $p_t\left(\bar{\Omega}(x_t)|\Omega(x_0) = y\right) \approx p_t\left(\left[\bar{\Omega}(x_t); \Omega(x_t)\right]\right)$, where $\hat{\Omega}(x_t)$ denotes samples from $p_t\left(\Omega(x_t)|\Omega(x_0 = y)\right)$, and $\left[\bar{\Omega}(x_t); \Omega(x_t)\right]$ represents the concatenation of two sets of dimensions. More concretely, in their approach, the samples for $x_t^{\mathrm{ob}}$ are replaced by exact samples from the forward process $q(x_t^{\mathrm{ob}}|x^{\mathrm{ob}})$ in Equation 14, at each iteration, while the sampling procedure for updating $x_t^{\mathrm{ta}}$ is still sampling from $p_\theta(x_t|x_{t+1})$. The samples $x_t^{\mathrm{ob}}$ then have the correct marginal distribution, and $x_t^{\mathrm{ta}}$ will conform with $x_t^{\mathrm{ob}}$ through the denoising process. Using this strategy, we can generate an intact sample that follows the correct conditional distribution in addition to the correct

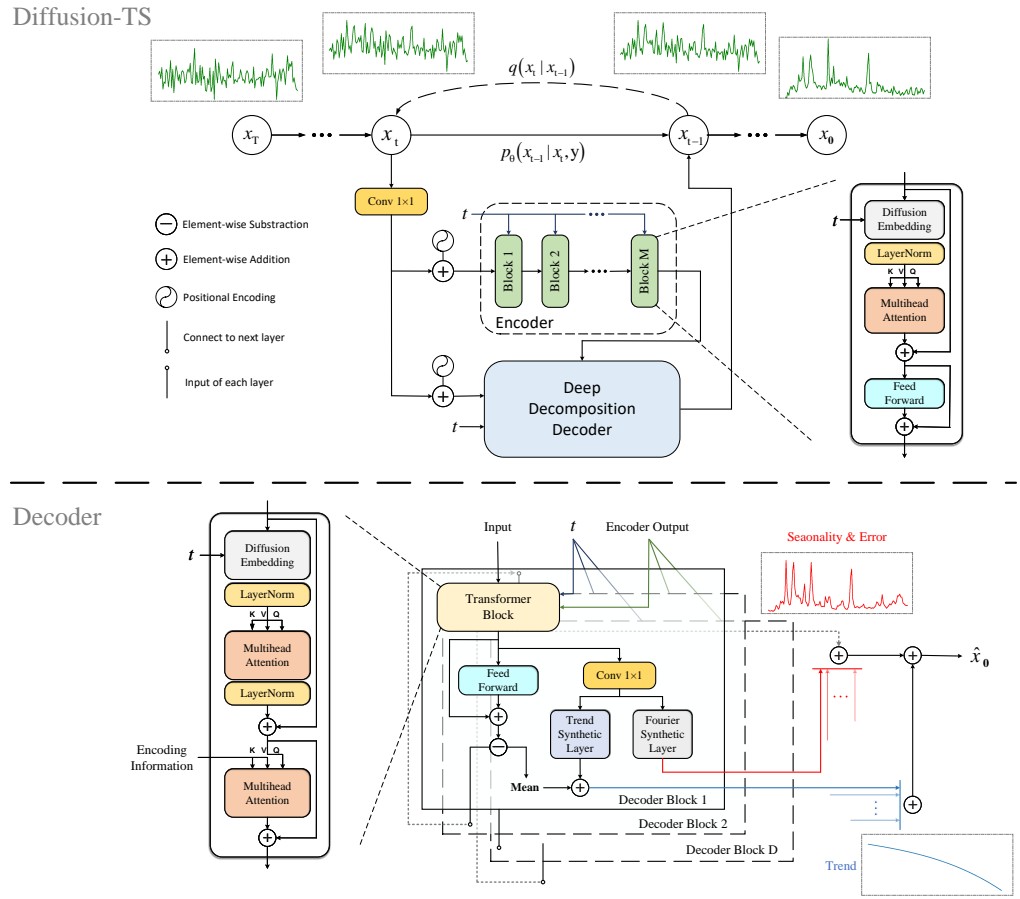

Figure 13: Overview figure of the Diffusion-TS using a Transfomer inspired architecture with the decomposition design.

marginal. We will refer to the infilling method for conditional sampling as replace-based imputation with a diffusion model.

## E   MODEL DETAILS

We now describe the details about our model architecture. The overview of Diffusion-TS is shown in Figure 13. As described in main paper, the framework contains two parts: a sequence encoder and an interpretable decoder. Each encoder block contains a full attention and a feed forward network block, while each transformer block in the decoder consists of a full attention and a cross attention to combine the encoding information. For the diffusion embedding, we follow previous works (Ho et al., 2020; Gu et al., 2022) with transformer sinusoidal positional embedding to encode the diffusion step. The diffusion step $t$ is injected into the network using the Adaptive Layer Normalization operator, which can be written as: $a_t \text{Laynorm}(w) + b_t$ where $w$ is the intermediate activations, $a_t$ and $b_t$ are obtained from a linear projection of the diffusion embedding.

# F ALGORITHMS

The full reconstruction-guided sampling algorithm is illustrated on the left of F, where $\text{Replace}(\cdot)$ denotes replace-based imputation algorithm introduced in D. As aforementioned, we take multiple gradient steps for each diffusion step to improve the generative quality. Note that the reverse process of a diffusion model can be viewed as two periods: creating in the early stages and smoothing in the rest stages, we thus run more gradient updates at large diffusion step $t$ to guide the generation then reduce the number of gradient steps in stages to accelerate sampling. The algorithm in the right of F presents this optimization in detail.

---

**Algorithm 1** Reconstruction-guided Sampling

**Require:** gradient scale $\eta$, conditional data $x_a$, trade-off coefficient $\gamma$
1:  $x_T \sim \mathcal{N}(0, \mathbf{I})$;
2:  **for** all $t$ from $T$ to 1 **do**
3:     $[\hat{x}_a, \hat{x}_b] \leftarrow \hat{x}_0(x_t, t, \theta)$;
4:     $\mathcal{L}_1 = \|x_a - \hat{x}_a\|_2^2$;
5:     $x_{t-1} \leftarrow \mathcal{N}(\mu(\hat{x}_0(x_t, t, \theta), x_t), \Sigma)$;
6:     $\mathcal{L}_2 = \|x_{t-1} - \mu(\hat{x}_0(x_t, t, \theta), x_t)\|_2^2 / \Sigma$;
7:     $\tilde{x}_0 = \hat{x}_0(x_t, t, \theta) + \eta \nabla_{x_t}(\mathcal{L}_1 + \gamma \mathcal{L}_2)$;
8:     $x_{t-1} \leftarrow \mathcal{N}(\mu(\tilde{x}_0, x_t), \Sigma)$;
9:     $x_{t-1} \leftarrow \text{Replace}(x_a, x_{t-1}, t)$;
10: **end for**

---

**Algorithm 2** Optimized Conditional Sampling

**Require:** gradient scale $\eta$, conditional data $x_a$, trade-off coefficient $\gamma$, times set $K$
1:  $x_T \sim \mathcal{N}(0, \mathbf{I})$;
2:  **for** all $t$ from $T$ to 1 **do**
3:     **for** all $i$ from $K[t]$ to 1 **do**
4:        $[\hat{x}_a, \hat{x}_b] \leftarrow \hat{x}_0(x_t, t, \theta)$;
5:        $\mathcal{L}_1 = \|x_a - \hat{x}_a\|_2^2$;
6:        $x_{t-1} \leftarrow \mathcal{N}(\mu(\hat{x}_0(x_t, t, \theta), x_t), \Sigma)$;
7:        $\mathcal{L}_2 =$
              $\|x_{t-1} - \mu(\hat{x}_0(x_t, t, \theta), x_t)\|_2^2 / \Sigma$;
8:        $x_t = x_t + \eta \nabla_{x_t}(\mathcal{L}_1 + \gamma \mathcal{L}_2)$;
9:     **end for**
10:    $x_{t-1} \leftarrow \mathcal{N}(\mu(\hat{x}_0(x_t, t, \theta), x_t), \Sigma)$;
11:    $x_{t-1} \leftarrow \text{Replace}(x_a, x_{t-1}, t)$;
12: **end for**

---

# G EXPERIMENT DETAILS

## G.1 DATASETS

Table 11 shows the statistics of the datasets and all datasets are available online via the link.

Table 11: Dataset Details.

| Dataset | # of Samples | dim | Link |
|---------|--------------|-----|------|
| Sines   | 10000        | 5   | https://github.com/jsyoon0823/TimeGAN |
| Stocks  | 3773         | 6   | https://finance.yahoo.com/quote/GOOG |
| ETTh(1) | 17420        | 7   | https://github.com/zhouhaoyi/ETDataset |
| MuJoCo  | 10000        | 14  | https://github.com/deepmind/dm_control |
| Energy  | 19711        | 28  | https://archive.ics.uci.edu/ml/datasets |
| fMRI    | 10000        | 50  | https://www.fmrib.ox.ac.uk/datasets |

## G.2 BASELINES

We apply and optimize the following accessible source codes for generative experiments.

- TimeGAN: https://github.com/jsyoon0823/TimeGAN
- TimeVAE: https://github.com/abudesai/timeVAE
- Cot-GAN: https://github.com/tianlinxu312/cot-gan
- Diffwave: https://diffwave-demo.github.io/
- CSDI / DiffTime: https://github.com/ermongroup/CSDI

For GAN-based methods, we use the 3-layer GRU-based neural network architecture with a hidden size that is 4 times larger than the feature size. We modify settings of TimeVAE and Diffwave so that they have roughly the same trainable parameter size as ours. For CSDI, we set the number of residual layers as 4, residual channels as 64, and attention heads as 8. We also change the kernel-size of the convolutional layers following the settings of DiffTime. Finally, both Diffwave and CSDI use the same diffusion settings (e.g. diffusion steps) as ours if necessary.

### G.3 EVALUATION METRICS

**Discriminative** & **Predictive score.** The discriminative score is calculated as $|\text{accuracy} - 0.5|$, while the predictive score is the mean absolute error (MAE) evaluated between the predicted values and the ground-truth values in test data. For a fair comparison, we reuse the experimental settings of TimeGAN Yoon et al. (2019) for the discriminative and predictive score. Both the classifier and sequence-prediction model use a 2-layer GRU-based neural network architecture.

**Context-FID score.** A lower FID score means the synthetic sequences are distributed closer to the original data. Paul et al. (2022) proposed a Frechet Inception distance (FID)-like score, Context-FID (Context-Frechet Inception distance) score by replacing the Inception model of the original FID with a time series representation learning method called TS2Vec (Yue et al., 2022). They have shown that the lowest scoring models correspond to the best-performing models in downstream tasks and that the Context-FID score correlates with the downstream forecasting performance of the generative model. Specifically, we first sample synthetic time series and real-time series respectively. Then we compute the FID score of the representation after encoding them with a pre-trained TS2Vec model.

**Correlational score.** Following Ni et al. (2020), we estimate the covariance of the $i^{th}$ and $j^{th}$ feature of time series as follows:

$$\text{cov}_{i,j} = \frac{1}{T}\sum_{t=1}^{T} X_i^t X_i^t - \left(\frac{1}{T}\sum_{t=1}^{T} X_i^t\right)\left(\frac{1}{T}\sum_{t=1}^{T} X_j^t\right). \tag{22}$$

Then the metric on the correlation between the real data and synthetic data is computed by

$$\frac{1}{10}\sum_{i,j}^{d} \left| \frac{\text{cov}_{i,j}^r}{\sqrt{\text{cov}_{i,i}^r \text{cov}_{j,j}^r}} - \frac{\text{cov}_{i,j}^f}{\sqrt{\text{cov}_{i,i}^f \text{cov}_{j,j}^f}} \right|, \tag{23}$$

## H ADDITIONAL VISUALIZATIONS

We introduce additional visualization and distribution outcomes in Figure 14 and Figure 15.

## I ADDITIONAL EXAMPLES

We present additional examples to illustrate various imputation examples to show the characteristic of time-series imputation and forecasting on the Energy dataset in Figures 16 to 19.

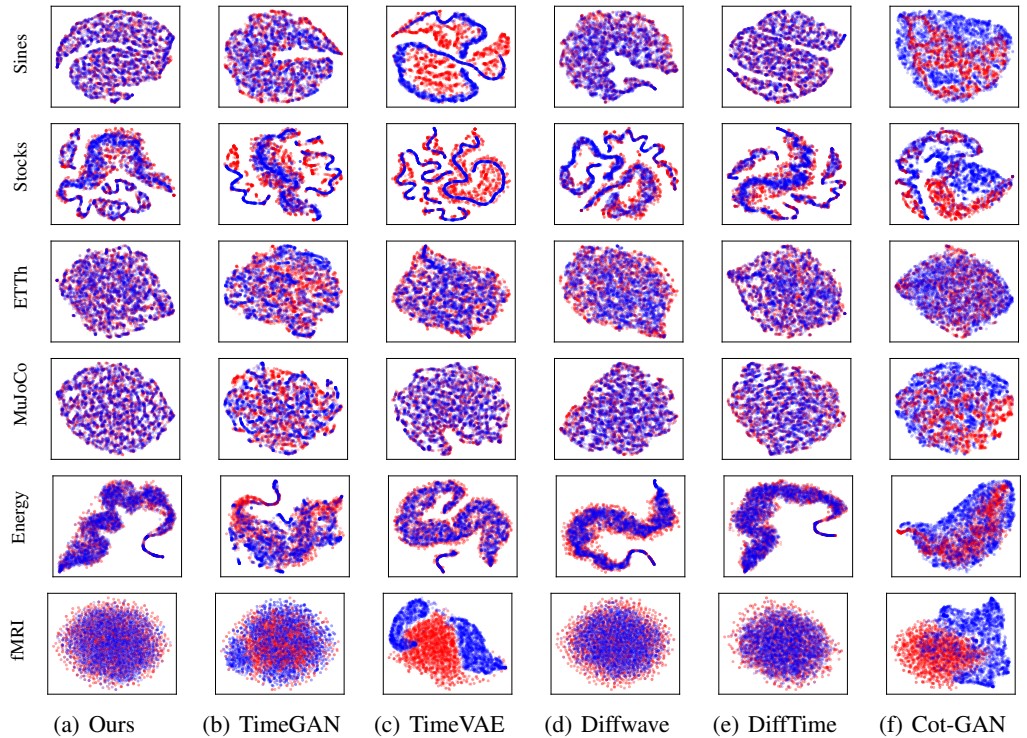

Figure 14: t-SNE visualizations of all methods on multivariate datasets. Red is for original data, and blue for synthetic data.

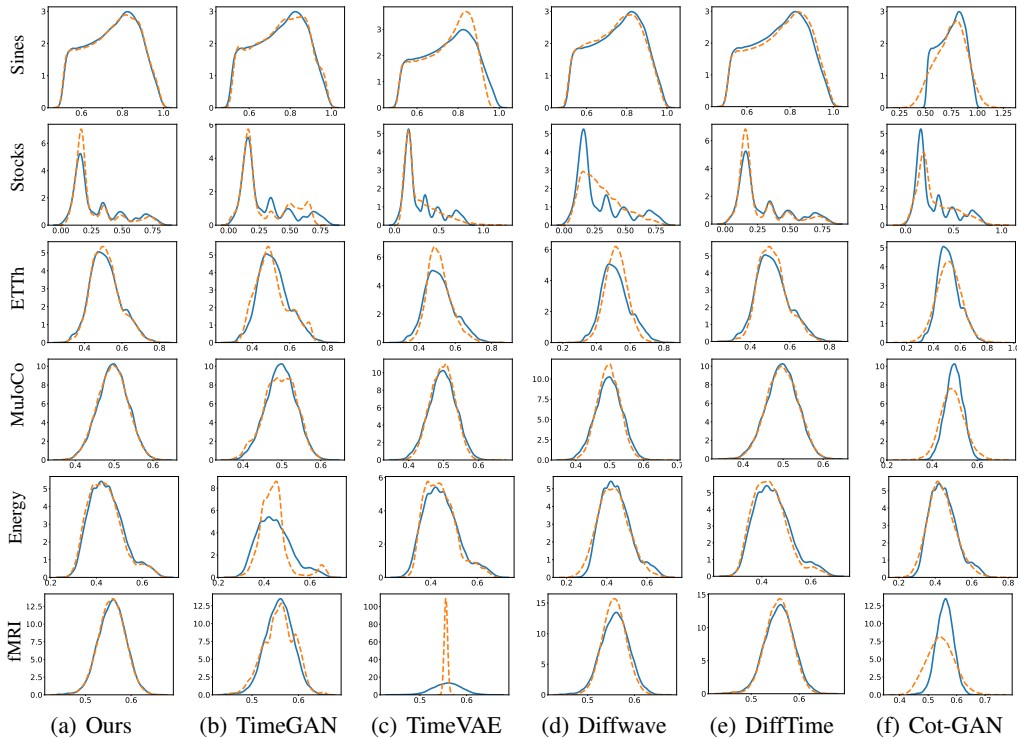

Figure 15: Distributions of all methods on multivariate datasets. blue solid line is for original data, and yellow dotted line for synthetic data.

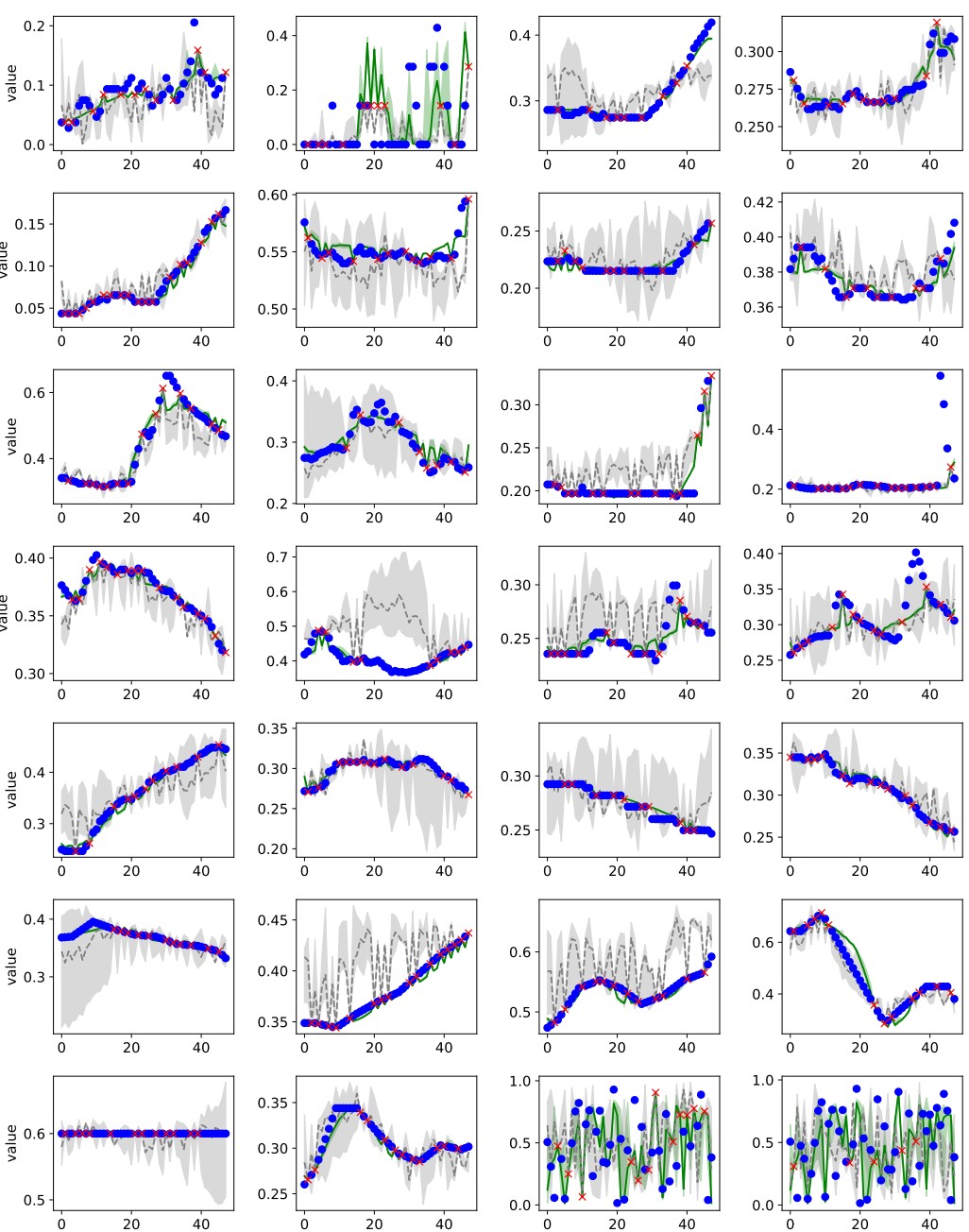

Figure 16: Comparison of imputation for the Energy dataset (90% missing). The result is for a time series sample with all 28 features. The red crosses show observed values and the blue circles show ground-truth imputation targets. Green and gray colors correspond to Diffusion-TS and Diffwave, respectively. For each method, median values of imputations are shown as the line and 5% and 95% quantiles are shown as the shade.

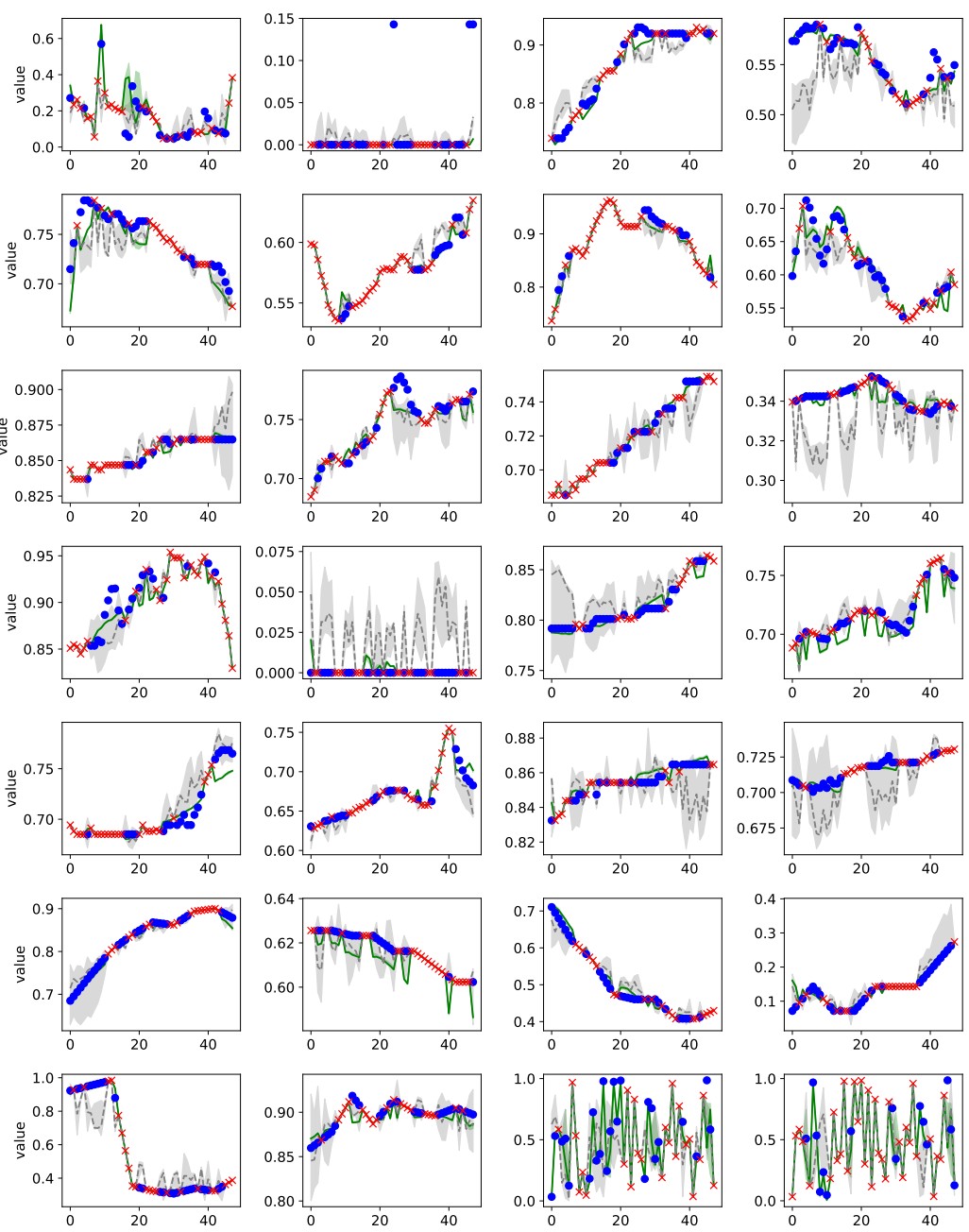

Figure 17: Comparison of imputation for the Energy dataset (50% missing). The result is for a time series sample with all 35 features. The red crosses show observed values and the blue circles show ground-truth imputation targets. Green and gray colors correspond to Diffusion-TS and Diffwave, respectively. For each method, median values of imputations are shown as the line and 5% and 95% quantiles are shown as the shade.

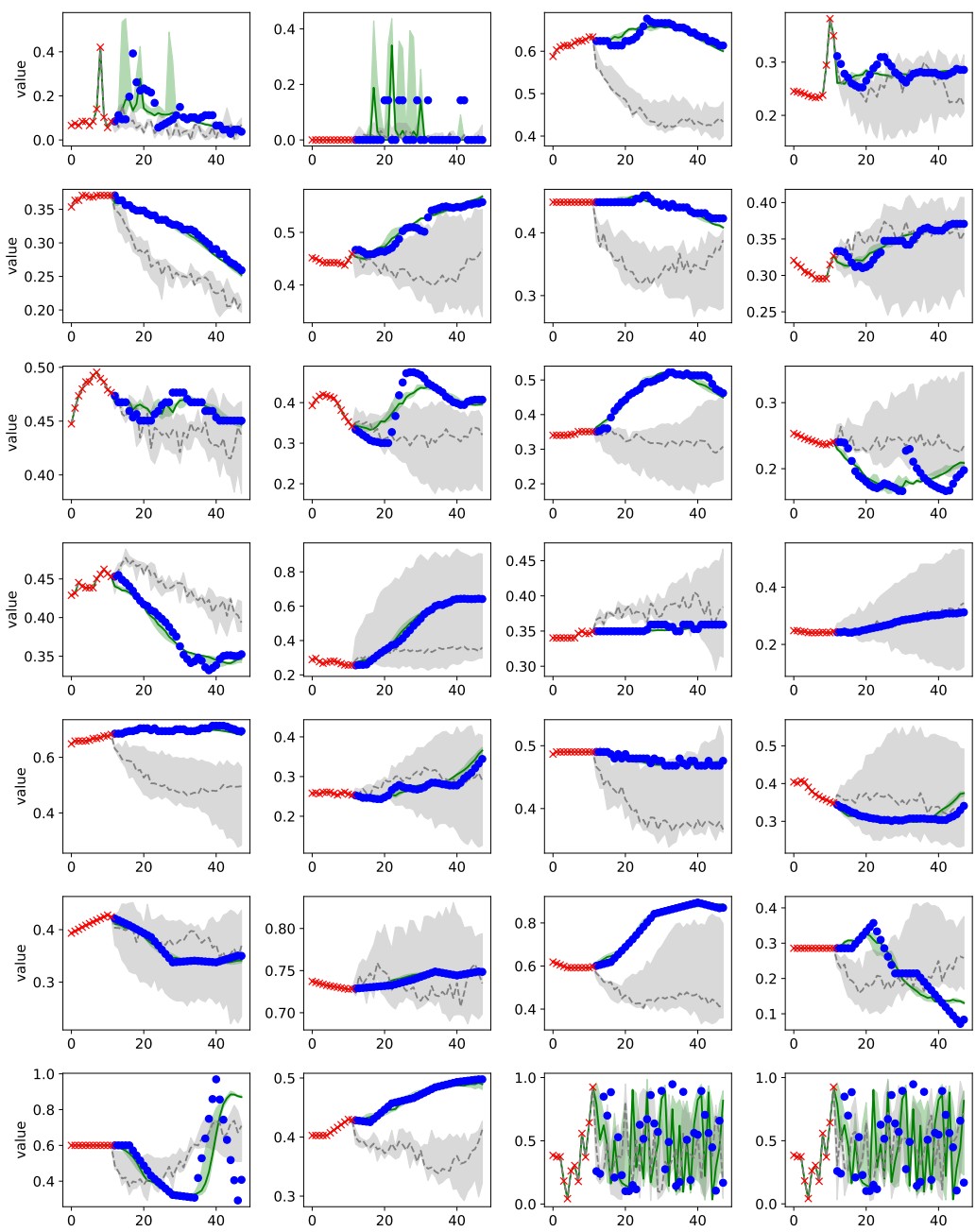

Figure 18: Comparison of forecasting for the Energy dataset (36 forecasting window). The result is for a time series sample with all 28 features. The red crosses show observed values and the blue circles show ground-truth imputation targets. Green and gray colors correspond to Diffusion-TS and Diffwave, respectively. For each method, median values of imputations are shown as the line and 5% and 95% quantiles are shown as the shade.

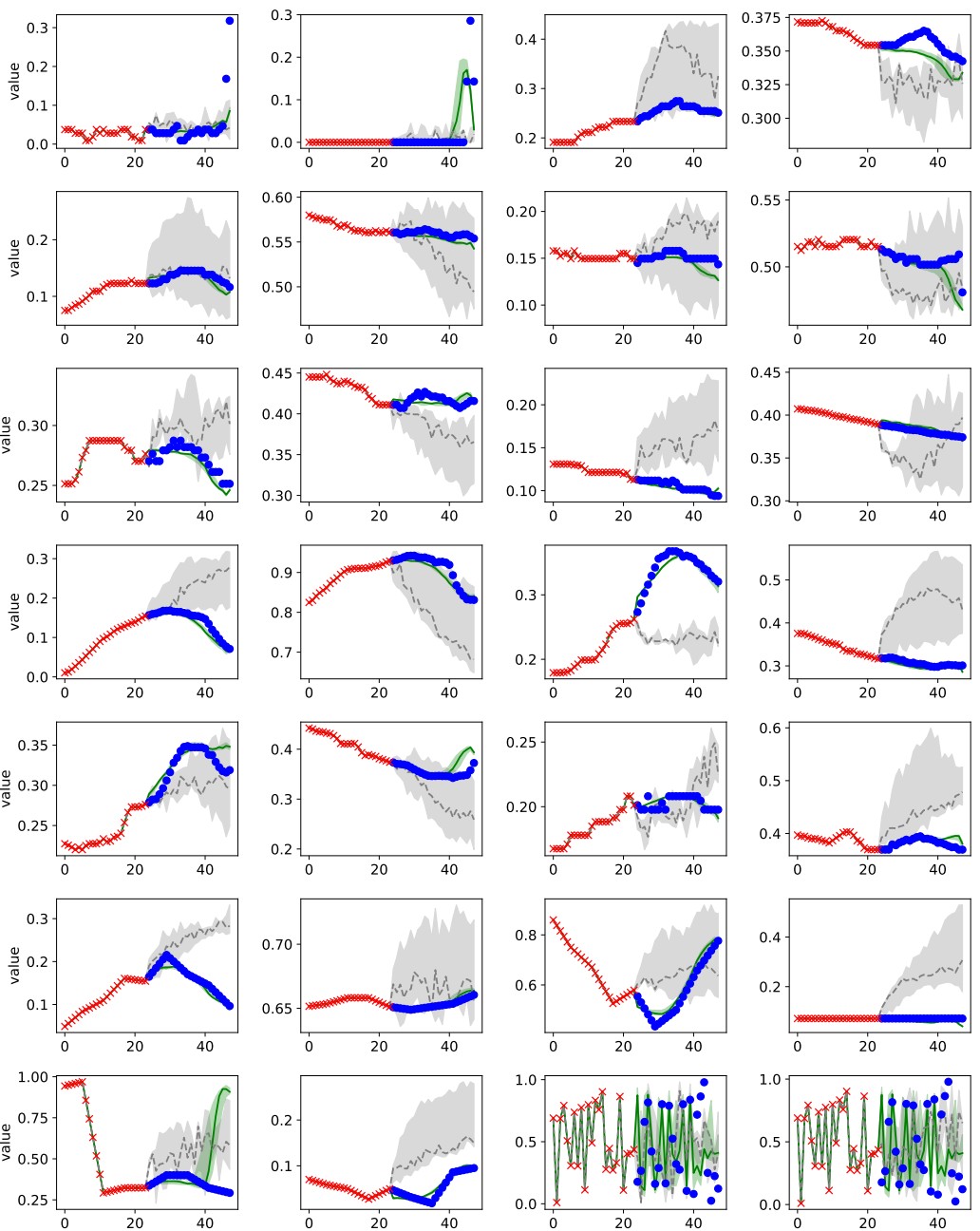

Figure 19: Comparison of forecasting for the Energy dataset (24 forecasting window). The result is for a time series sample with all 28 features. The red crosses show observed values and the blue circles show ground-truth imputation targets. Green and gray colors correspond to Diffusion-TS and Diffwave, respectively. For each method, median values of imputations are shown as the line and 5% and 95% quantiles are shown as the shade.

