# OpenReview forum: "Diffusion-TS: Interpretable Diffusion for General Time Series Generation"
_ICLR.cc/2024/Conference — ICLR 2024 poster_

### Official Review · Reviewer_iyrP · 2023-10-30

**Soundness:** 2 fair
**Presentation:** 3 good
**Contribution:** 2 fair
**Rating:** 6
**Confidence:** 3

**Summary:**

The authors propose Diffusion-TS, a diffusion method for multivariate time series generation. The method involves an interpretable decomposition into trend and seasonality components. The diffusion model is trained using a joint L2 and Fourier loss, and the authors apply conditional generation techniques to apply the method to forecasting and imputation directly without changes in the model. Experiments show Diffusion-TS achieves state-of-the-art performance on time series datasets, including in an irregularly sampled setting.

**Strengths:**

Originality:
- As far as the reviewer is aware, the authors are the first to combine a trend-seasonality decomposition technique with diffusion for time series generation. The authors mention this choice leads to improved performance across the various evaluation metrics.

Quality:
- The experiments are generally well-constructed. A variety of datasets and evaluation metrics are used for standard unconditional generation, and several ablation studies are included in the appendix.
- The authors show qualitative results showing the trend and seasonality components predicted by the model generally behave as intended.

Clarity:
- The paper is generally well-written, and the experiments are described clearly.

Significance:
- The experimental results are quite compelling. In the unconditional time series generation settings (Table 1), Diffusion-TS outperforms existing methods. The performance gap is especially clear for higher-dimensional datasets and long-term generation (Table 2).

**Weaknesses:**

- Generally, it's a bit unclear to the reviewer which design choices are crucial to the improvement in performance, especially the trend-seasonality decomposition. Two detailed techniques are involved in implementing the decomposition, the polynomial regressor and the Fourier synthetic layers, but it's unclear to what extent these techniques contribute to the model performance.
- It's a bit unclear how compelling the qualitative results are. For example, in Figure 3, the trend components seem quite uninformative compared to the season/error components. In Figure 4, it's unclear whether the t-SNE visualizations support the claim that the Diffusion-TS distribution better aligns with the data distribution than TimeGAN distribution.
- Experimental metrics reported are inconsistent. For example, correlational and context-FID scores are reported in Table 2, and discriminative and predictive scores are reported in Figure 7 and Appendix C.1/C.2.

**Questions:**

Main:
- In Appendix C.2, what is the "w/o interpretability" model? Is this equivalent to $x_0(x_t, t, \theta) = R$ in Eqn 7, setting the $V_{tr}^t$ and $S_{i, t}$ terms to $0$? How crucial are the polynomial regressor and Fourier synthetic layer techniques?
- How well does the trend-seasonality decomposition method of Diffusion-TS achieve what's intended for the toy model (Eqn 1), for example on synthetic data?
- How do discriminative and predictive scores compare on the long-term time series generation task (Table 2)?
- For the long-term generation task (Table 2), why is it that Diffusion-TS seems to perform generally better with longer time series? It seems like longer time series generation should be a more difficult task.
- In Figure 6, is MAE computed only over missing data (imputation targets) or over the full time series including existing data? For Diffusion-TS-G, since a soft constraint is used to enforce the conditional generation, how closely do the generated time series for imputation/forecasting match the existing data?
- How does the number of parameters compare between Diffusion-TS and its competitors?

Clarifications:
- How is the data in Figure 3 generated? Is column b the ground truth and column a the result of adding noise?

---

> ### Author Response · Authors · 2023-11-20
> **Reponse to Reviewer iyrP (Part 1)**
>
> We thank the reviewer for the comments. We address individual concerns below.
>
> > Generally, it's a bit unclear to the reviewer which design choices are crucial to the improvement in performance, especially the trend-seasonality decomposition. How crucial are the polynomial regressor and Fourier synthetic layer techniques?
>
> Table 9 in Appendix C.7 has reported the ablations for the components of trend & seasonality and Fourier regularization on 24-length time series. When the dataset, i.e. fMRI, has a high frequency and dimension, a network with interpretable design achieves the most significant performance improvement. For validating a more explict improvement, we added the ablation study in long sequence infilling on MuJoCo dataset to demonstrate that each disentanglement plays an important role in improving the performance on generative tasks. (This has been included as Appendix C.7 of the revised paper)
>
> |Model|$70 \%$ Missing|$80 \%$ Missing|$90 \%$ Missing|
> |:--:|:--:|:--:|:--:|
> |Residual|0.51(1)|0.59(7)|0.85(10)|
> |Residual+Season|0.45(5)|0.52(3)|0.77(9)|
> |Residual+Trend|0.46(2)|0.50(5)|0.80(7)|
> |Season+Trend|0.63(3)|1.05(6)|1.42(10)|
> |Diffusion-TS|__0.37(3)__|__0.43(3)__|__0.73(12)__|
>
> > In Figure 3, the trend components seem quite uninformative compared to the season/error components.
>
> First, the constraints imposed by polynomial regressor are very strong to ensure slow-varying trend components. The same phenomenon can be found in the experimental part of NBEATS [1]. Second, juggling various noisy data in diffusion models exacerbates the information deficit. In our early experiments, we also tried to use moving average to model trends like Autoformer [2] and Fedformer [3], but this leads to worse quality of generated samples. However, the ''uninformative'' curve does not hurt since we are only considering coarse-grained trends.
>
> > How well does the trend-seasonality decomposition method of Diffusion-TS achieve what's intended for the toy model (Eqn 1), for example on synthetic data?
>
> Thank you very much for your comments. To better validate the toy model, we have added an extra interpretability study on synthetic data in Appendix C.5 of the revised paper. Through experiments, we can clearly find that the learned disentangled components are very similar to the ground truth. This verifies the interpretability of our proposed method.
>
> > In Figure 4, it's unclear whether the t-SNE visualizations support the claim that the Diffusion-TS distribution better aligns with the data distribution than TimeGAN distribution.
>
> Actually in Figure 4, we can see that the 2-dimensional points from our generated samples are distributed more evenly than that of TimeGAN’s samples. That means the samples generated by our method more coincide with the ground truth. To better visualize the superiority of our method, we  also added the PCA and t-SNE visualizations of long-term series data on ETTh data set in the Appendix C.2 of the revised paper. In these figures, the synthetic samples generated by Diffusion-TS show significant superiority than TimeGAN.
>
> > Experimental metrics reported are inconsistent.
>
> In the revised paper, we added the results on the complete metrics in the Appendix C.1. We also supplemented the 24-sequence ablation experiment in the Appendix C.7 to ensure the consistency of experimental metrics.
>
> > In Appendix C.2, what is the "w/o interpretability" model? Is this equivalent to $R$, setting the $V_{tr}^t$ and $S_{i,t}$?
>
> Yes, the "w/o interpretability" model remove both the polynomial regressor and Fourier synthetic layer.
>
> > For the long-term generation task (Table 2), why is it that Diffusion-TS seems to perform generally better with longer time series? It seems like longer time series generation should be a more difficult task.
>
> Thank you for raising this question. First of all, the self-attention mechanism of Transformer naturally supports long sequence modeling. The similar results that the attention-based models perform better in longer time series generation were also reported in PSA-GAN [4]. In addition, the trend-seasonal decomposition applied in our framework is more suitable for processing long time series, as both trend and seasonality are long-term time properties.
>
> references:
>
> [1] N-beats: Neural basis expansion analysis for interpretable time series forecasting, 2020.
>
> [2] Autoformer: Decomposition transformers with auto-correlation for long-term series forecasting. Advances in Neural Information Processing Systems, 34:22419–22430, 2021.
>
> [3] Fedformer: Frequency enhanced decomposed transformer for long-term series forecasting, 2022.
>
> [4] Psa-gan: Progressive self attention gans for synthetic time series, 2022.

---

> > ### Author Response · Authors · 2023-11-20
> > **Reponse to Reviewer iyrP (Part 2)**
> >
> > > In Figure 6, is MAE computed only over missing data (imputation targets) or over the full time series including existing data? How closely do the generated time series for imputation/forecasting match the existing data?
> >
> > First, we apologize for mistakenly writing MSE as MAE in Figure 6. We have corrected this typo in the revised paper. The MSE here is computed only over missing data (imputation targets). To better visualize how well the generated time series match the existing data, we added example prediction results on Solar data set in Appendix C.3, although we have listed many imputation/forecasting examples at the end of the appendix.
> >
> > > How does the number of parameters compare between Diffusion-TS and its competitors?
> >
> > The model size of our method and baselines are listed in the table below. Diffusion models have generally more parameters than other methods. Nevertheless, our Diffusion-TS model has much fewer parameters than Diffwave, due to. This table has been included as Appendix C.6 of the revised paper.
> >
> > |Model|Sines|Stocks|Energy|
> > |:--:|:--:|:--:|:--:|
> > |TimeVAE|97,525|104,412|677,418|
> > |TimeGAN|34,026|48,775|1,043,179|
> > |Cot-GAN|40,133|52,675|601,539|
> > |Diffwave|533,592|599,448|1,337,752|
> > |Diffusion-TS|232,177|291,318|1,135,144|
> >
> > > How is the data in Figure 3 generated? Is column b the ground truth and column a the result of adding noise?
> >
> > Yes, the noisy data (in column (a)) is the input after adding 50 steps of noise (500 steps in total) to the ground truth (in column (b)). Then we take out the output of each module: Trend
> > Synthetic Layers as Trend (in column (d)), and Fourier Synthetic Layers (along with the output of the last block) as Season & Error ((in column (e))). Finally, we report their sum in column (c).

---

> > > ### Comment · Reviewer_iyrP · 2023-11-23
> > >
> > > Thanks for the detailed response. The additional details and experiments are compelling and help clarify the role of the season/trend components. Therefore I'm willing to raise my score to 6.

---

> > > > ### Author Response · Authors · 2023-11-23
> > > > **Official Comment by Authors**
> > > >
> > > > We would like to sincerely thank the reviewer for re-evaluating our manuscript and for the constructive feedback. We are pleased to see that our responses and clarifications have positively impacted the assessment of our work.

---

### Official Review · Reviewer_Prro · 2023-11-03

**Soundness:** 2 fair
**Presentation:** 2 fair
**Contribution:** 2 fair
**Rating:** 5
**Confidence:** 4

**Summary:**

This paper proposes a diffusion model for time series generation. The model involves a custom architecture with components that help synthesizing the trend and seasonal components of the time series. The model is trained to directly estimate the observation from arbitrary diffusion steps. The objective function of the diffusion model is also modified to incorporate a loss based on fourier transform of the time series. Experiments show improved performance over existing generative models in terms of different generative metrics.

**Strengths:**

- The model explores both unconditional and conditional generation using a single model.
- The model performs better than baseline models in terms of time series generation.

**Weaknesses:**

- The claim "little attention has been given to leveraging the powerful generative ability for general time series production" is not entirely correct. See [1] and [2], for example. Furthermore, in the light of [2], the claim "... **first** DDPM-based framework for both unconditional and conditional time series synthesis" is not correct either. [2] proposes a self-guidance mechanism to use unconditional diffusion models for conditional time series tasks and also studies the unconditional generative properties of the model. This limits the novelty of the paper and Sec. 3.4, in particular.
- The paper is poorly written and does not tell a coherent story. Sec. 3.2 reads like an arbitrary combination of ideas. The individual components are also not analyzed later via ablations. The manuscript also has many typographical errors (synthetic instead of synthesis, DSDI instead of CSDI).
- The evaluation on the conditional tasks is limited. The model is only compared against Diffwave and CSDI (which is fairly close to Diffwave) and baselines from time series forecasting literature are missing. It is also unclear how these CSDI and Diffwave baselines were trained.



[1] Lim, Haksoo, et al. "Regular Time-series Generation using SGM." arXiv preprint arXiv:2301.08518 (2023).
[2] Kollovieh, Marcel, et al. "Predict, refine, synthesize: Self-guiding diffusion models for probabilistic time series forecasting." arXiv preprint arXiv:2307.11494 (2023).

**Questions:**

- What is the empirical significance of the trend & seasonality synthesis blocks and the fourier regularization? Did the authors conduct ablations for these components? (Check the results in [1] which doesn't use any of these components)
- Why does Diffwave perform so much worse in the case of long time series generation? The sequence lengths considered in this work are smaller than those used in audio synthesis. Is it possible that there is a bug in the experiments setup?
    - Why did not authors not compare against a CSDI-like model for unconditional generation? That would probably be a better comparison and ablation for the components proposed in this work.

[1] Lim, Haksoo, et al. "Regular Time-series Generation using SGM." arXiv preprint arXiv:2301.08518 (2023).

---

> ### Author Response · Authors · 2023-11-20
> **Reponse to Reviewer Prro (Part 1)**
>
> We thank the reviewer for the comments. We address individual concerns below.
>
> > The claim "little attention has been given to leveraging the powerful generative ability for general time series production" is not entirely correct. The claim "... first DDPM-based framework for both unconditional and conditional time series synthesis" is not correct either.
>
> We apologize for the unprecise expression. Thanks for the reviewer for providing more concurrent work. We modified the introduction and related work in the revised paper to embrace these work, and highlighted the differences between the recent proposed DDPM-based framework and the work proposed in our paper. Particularly, our method is expressly focused on the problem of __general__ time series generation, which not only represents (un)conditional time series synthesis, but also represents high (low) dimension and long (short) time series synthesis. For the framework proposed in [1], it focused on univariate time series using a modification of DiffWave with S4 layers, which was designed for single-channel audio data. While the framework proposed in [2] was only able to generate unconditional time series of length 24, and it is hardly to extended to long-term series generation Besides, it was quite resource-intensive. For example, it requires 3318.99s (255 times of ours: 13s) for sampling 1000 Stock sequences and 1620.84s (49 times of ours: 33s) for Energy sequences.
>
> > The paper is poorly written and does not tell a coherent story. Sec. 3.2 reads like an arbitrary combination of ideas. The manuscript also has many typographical errors (synthetic instead of synthesis, DSDI instead of CSDI).
>
> Thanks very much for pointing out this issue.. We have improved the organization and fixed typographical errors to enhance the readability of our paper. In the introduction, we present more motivations and highlight several aspects in which our decomposition architecture performs more favorably in terms of interpretability and effectiveness We reorganized Section 3, especially Section 3.2, to emphasize the rationality of our ideas.
>
> > The evaluation on the conditional tasks is limited.
>
> We agree that comparing our method to additional baselines on the conditional tasks will further confirm that our approach is relevant in practice. To demonstrate that the performance of Diffusion-TS, we repeat time series imputation and forecasting experiments in SSSD [3]. We report an averaged MSE for a single imputation per sample on the MuJoCo data set of length 100. The results are shown in table below.
>
> |Model|$70 \%$ Missing|$80 \%$ Missing|$90 \%$ Missing|
> |:--:|:--:|:--:|:--:|
> |RNN GRU-D|11.34|14.21|19.68|
> |ODE-RNN|9.86|12.09|16.47|
> |NeuralCDE|8.35|10.71|13.52|
> |Latent-ODE|3|2.95|3.6|
> |NAOMI|1.46|2.32|4.42|
> |NRTSI|0.63|1.22|4.06|
> |CSDI|__0.24(3)__|0.61(10)|4.84(2)|
> |SSSD|0.59(8)|1.00(5)|1.90(3)|
> |Diffusion-TS|0.37(3)|__0.43(3)__|__0.73(12)__|
>
> Then we test on the Solar data set collected from GluonTS [4], a forecasting task where the conditional values and forecast horizon are 168 and 24 time steps respectively. The results are shown in table below. We can see our model still performs well as it has the best performance against the baselines. (This has been included as Appendix C.3 of the revised paper).
>
> |Model|MSE|
> |:--:|:--:|
> |GP-copula|9.8e2±5.2e1|
> |TransMAF|9.30e2|
> |TLAE|6.8e2±7.5e1|
> |CSDI|9.0e2±6.1e1|
> |SSSD|5.03e2±1.06e1|
> |Diffusion-TS|__3.75e2±3.6e1__|
>
> References:
>
> [1] Predict, refine, synthesize: Self-guiding diffusion models for probabilistic
> time series forecasting. arXiv preprint arXiv:2307.11494, 2023.
>
> [2] Regular time-series generation using sgm. arXiv preprint arXiv:2301.08518, 2023.
>
> [3] Diffusion-based time series imputation and forecasting with structured state space models. arXiv preprint arXiv:2208.09399, 2022.
>
> [4] Gluonts: Probabilistic and neural time series modeling in python. The Journal of
> Machine Learning Research, 21(1):4629–4634, 2020.

---

> > ### Author Response · Authors · 2023-11-20
> > **Reponse to Reviewer Prro (Part 2)**
> >
> > > What is the empirical significance of the trend & seasonality synthesis blocks and the fourier regularization? Did the authors conduct ablations for these components?
> >
> > Table 9 in Appendix C.7 has reported the ablations for the components of trend & seasonality and Fourier regularization on 24-length time series. When the dataset, i.e. fMRI, has a high frequency and dimension, a network with interpretable design achieves the most significant performance improvement. For validating a more explict improvement, we added the ablation study in long sequence infilling on MuJoCo dataset to demonstrate that each disentanglement plays an important role in improving the performance on generative tasks. (This has been included as Appendix C.7 of the revised paper)
> >
> > |Model|$70 \%$ Missing|$80 \%$ Missing|$90 \%$ Missing|
> > |:--:|:--:|:--:|:--:|
> > |Residual|0.51(1)|0.59(7)|0.85(10)|
> > |Residual+Season|0.45(5)|0.52(3)|0.77(9)|
> > |Residual+Trend|0.46(2)|0.50(5)|0.80(7)|
> > |Season+Trend|0.63(3)|1.05(6)|1.42(10)|
> > |Diffusion-TS|__0.37(3)__|__0.43(3)__|__0.73(12)__|
> >
> > > It is also unclear how these CSDI and Diffwave baselines were trained.
> >
> > The training of Diffwave is same as uncondtional DDPMs, which predict the noise $\epsilon$ given a noisy sample $X_t$ at any step $t$. As for CSDI, we reuse the hyperparameter settings and scheme (we only apply noise to the portions of the time series to be imputed with training and sampling procedures) in the original paper, except that the training set used here is regular and side-information input is removed.
> >
> > > Why does Diffwave perform so much worse in the case of long time series generation? The sequence lengths considered in this work are smaller than those used in audio synthesis. Is it possible that there is a bug in the experiments setup?
> >
> > Regarding the baseline results, we ran all the models under the same environment for a fair comparison. We agree that the length of time series considered in our work are shorter than those used in speech synthesis. Nonetheless, we want to emphasize that both Context-FID score and Correlational score are affected by the spatial-temporal dependency of the sample. Since Diffwave is designed for the speech synthesis with single (or two) dimension(s), it does not consider the spatial correlations among the time series data. Hence, the samples generated by Diffwave do not have evident spatial dependency. Furthermore, results in CSDI also show that both temporal and feature correlations are important for multivariate time series modeling (the model used in CSDI is actually a collection of Diffwave and spatio-temporal attention). That being said, we have done our best to adjust DiffWave in order to evaluate on the popular Long Sequence Time-series benchmark. We thus believe such differences may be unsurprising due to the spatio-temporal correlations that become more complicated as dimensions and length increase.
> >
> > > Why did not authors not compare against a CSDI-like model for unconditional generation?
> >
> > Thank you for directing us to the paper. Since the original paper like CSDI did not discuss the unconditional generation task, we believe this comparison is unfair and has little significance in demonstrating our performance. However, we still conducted comparative experiments on three data sets as shown in the table below.
> >
> > |Model|Metirc|Sines|Stocks|Energy|
> > |:--:|:--:|:--:|:--:|:--:|
> > |CSDI-unconditional|Dis|0.013±.006|0.097±.016|0.445±.004|
> > |Diffusion-TS|Dis.|__0.006±.007__|__0.067±.015__|__0.122±.003__|
> > |CSDI-unconditional|Pre.|__0.093±.000__|0.038±.001|0.252±.000|
> > |Diffusion-TS|Pre.|__0.093±.000__|__0.036±.000__|__0.250±.000__|

---

> > > ### Comment · Reviewer_Prro · 2023-11-22
> > >
> > > Thank you for pointing out the table in the Appendix. It would be great if the discussion on the components can be improved in the main text.
> > >
> > > Thank you for results on CSDI. I don't think it's an unfair comparison. It's just another diffusion model that can also be trained unconditionally. The performance in terms on the predictive score is pretty close which begs the point about the contributions of this paper, e.g., in terms of the decomposition and the Fourier loss.

---

> > > > ### Author Response · Authors · 2023-11-23
> > > > **Official Comment by Authors**
> > > >
> > > > We would like to thank the reviewer for the time dedicated to read our rebuttal, and we are happy to answer any additional questions.
> > > >
> > > > __The writing is still imprecise.__
> > > >
> > > > We apologize for the lack of clarity in our previous updating. Thanks for pointing out the imprecise places, we have rewritten the corresponding paragraphs in the paper.
> > > >
> > > > __It would be great if the discussion on the components can be improved in the main text.__
> > > >
> > > > We thank the reviewer for the suggestions. We acknowledge that the paper presentation would be more clear if the ablation experiments were improved to the main text. We add an ablation study subsection to the experimental part of the paper. However, due to the limited space, we only report discriminative score and predictive score and the detailed description remains in the appendix.
> > > >
> > > > __The performance in terms on the predictive score is pretty close.__
> > > >
> > > > We agree with the reviewer that CSDI and our method are comparable on unconditional generation. However, we still want to clarify that it is not easy to win a significant improvement on predictive score, and the small gap in the predictive score cannot be ignored. Because the results obtained using real data are 0.094±.001, 0.036±.001 and 0.250±.003 respectively, which means the scores achieved by our methods and the baseline methods were almost the optimal values. Nevertheless, we still believe that as the sequence length and complexity increase, the superiority of our proposed method in predicting score will become more significant.
> > > >
> > > > Since the remaining time for discussion is too limited to run unconditional CSDI as a benchmark method for all experiments, especially the part of long sequences (two metrics proposed by TimeGAN using RNN are very time-consuming), we promise to complete it in the next few days and release more experimental results upon the acception.

---

> > ### Comment · Reviewer_Prro · 2023-11-22
> >
> > Thank you for updating the claims in the introduction. However, I do feel that the writing is still imprecise. For instance, "Meanwhile, the rare work on unconditional time-related synthesis with diffusion models (Kong et al., 2021; Kollovieh et al., 2023; Lim et al., 2023) still struggle in synthesizing high-dimension or long time series." lumps univariate and multivariate models together and "We propose Diffusion-TS, the first generative framework to combine seasonal-trend decomposition techniques with diffusion models" is too specific of a thing to claim. I hope these things can be improved in the revision.
> >
> > Thank you for the new results on the conditional tasks. They look promising.

---

> ### Comment · Reviewer_Prro · 2023-11-23
>
> Thank you for the update. I am raising my score to 5.
>
> Minor comment: What I meant by imprecise is saying "the rare work on unconditional time-related synthesis with diffusion models still **struggle in synthesizing high-dimension** (Kong et al., 2021; Kollovieh et al., 2023)" when these papers actually do not claim to be be applicable on high-dimenisonal time series. It would be more precise to say something like "focus on low-dimensional or univariate time series".

---

> > ### Author Response · Authors · 2023-11-23
> > **Official Comment by Authors**
> >
> > Thanks for the comment. We have addressed your comment in the updated version.

---

### Official Review · Reviewer_bNeL · 2023-11-06

**Soundness:** 3 good
**Presentation:** 3 good
**Contribution:** 3 good
**Rating:** 8
**Confidence:** 4

**Summary:**

A transformer based denoising diffusion probabilistic model, Diffusion-TS, is proposed for generating multivariate time series.
Representations from Oreshkin 2020, Desai 2021, De Livera 2011 and Woo 2022 are adopted for trend and seasonality. Under time series model (1) and Fourier-based loss term following similar work as Ho 2020 and Fons 2022, the \hat{x}_{0} is estimated directly for unconditional time series.
The proposed method Diffusion-TS is compared with four other models TimeGAN 2019, TimeVAE 2021, Diffwave 2021, and Cot-Gan 2020 under 6 datasets (stock price, electricity transformer, etc. The proposed method works better than the other four methods most frequently under the evaluation rules suggested by Yoon 2019, Paul 2022 and Ni 2020 especially under high-dimensional datasets.
Extension to conditional model by adding gradients, as well as simulation results, are presented.

**Strengths:**

This idea and algorithm of the proposed method are generally well presented. It’s compared with multiple recent time series generation methods and simulation results outperform these in most cases.

**Weaknesses:**

Potentially the proposed method works well or even better than other existing methods under typical seasonal time series data. Pending further exploration/confirmation how well the performance the proposed method can be under times series of real data with different unique features like big jump in stock price. Also wondering the hyper-parameter selection impact on the result and convergence speed.

**Questions:**

P4, Please define similarly as in Woo (2022) your \omega_seas or at least add definition by English.
P4, Equation (4), please either split the A and \Phi definition in two rows or write as a vector, so would not be confused as compound operators.
P5, Could you clarify the Figure 2, ‘Decoder Block 1’, ..., ‘Decoder Block N’ each refers to under the N samples (i=1, ... ,N) of time-series signals on P2 or something else considering the squares/arrows in that figure.
P17, Can the dataset names or links be more specific? There’re multiple datasets available under some links.
P6 and P18, Not questioning the vitality of your result, just want to have a better understanding of the performance to be expected under your method for other potential new runs:
How different Table 1 results can be if you did try other hyper-parameters?
Prediction under jump model can be hard. How well the performance of your model can be when there’s big jump in observed data? (The Google stock data used is from 2004 to 2019, the big changes in 2020 and 2022 not included. The electricity transformer temperature and blood oxygen related data might be typical stable time series. I’m not familiar with MujoCo data.)

---

> ### Author Response · Authors · 2023-11-20
> **Reponse to Reviewer bNeL**
>
> We thank the reviewer for the comments. We address individual concerns below.
>
> > Please define similarly as in Woo (2022) your \omega_seas or at least add definition by English. Equation (4), please either split the A and \Phi definition in two rows or write as a vector, so would not be confused as compound operators.
>
> We thank the reviewer for the suggestion. We have moved the definition of $w_{(seas)}^{i,t}$ and $w_{(tr)}^{i,t}$ before the discussion in Section 3.2, and split Equation 4 to two equations.
>
> > Could you clarify the Figure 2, ‘Decoder Block 1’, ..., ‘Decoder Block N’ each refers to under the N samples (i=1, ... ,N) of time-series signals on P2 or something else considering the squares/arrows in that figure.
>
> We apologize for the lack of clarity in the description of the generative method. Here, N in Figure 2 only represents the number of residual blocks of deep decomposition architecture in the decoder. Therefore, there is no relationship between it and the number of samples appearing on P2. To avoid ambiguity, we change N to another letter K in the figure to avoid the confusion.
>
> > Can the dataset names or links be more specific? There’re multiple datasets available under some links.
>
> Since Sines and MuJoCo are synthetic data sets, we only provide the link to the source code. Additionally, we corrected the dataset link for ETTh data set.
>
> > Wondering the hyper-parameter selection impact on the result and convergence speed.
>
> We did limited hyperparameter tuning in this study to find default hyperparemters that perform
> well across datasets. Due to the time limitation, we were not able to repeat all experiments multiple times. To better illustrate, we added an extra hyperparameter tuning study in Appendix C.6 of the revised paper and will release more experimental results upon the acception. Furthermore, we have also added the impact of the scale parameter $\gamma$ in that subsection:
>
> |$\gamma$|$70 \%$ Missing|$80 \%$ Missing|$90 \%$ Missing|
> |:--:|:--:|:--:|:--:|
> |1.|2.8(13)|4.1(10)|6.8(17)|
> |1e-1|__0.37(4)__|0.45(0)|0.82(9)|
> |5e-2|__0.37(3)__|__0.43(3)__|__0.73(12)__|
> |1e-2|0.60(5)|0.70(10)|1.07(14)|
> |1e-3|3.1(8)|7.2(20)|19.6(22)|
>
> With perceptual qualities superior to GANs while avoiding the optimization challenges of adversarial training, the convergence speed of the diffusion model is not greatly affected by the hyper-parameters. And we compare the time to train Diffusion-TS with TimeGAN in the table below.
>
> |Model|Sines|Stocks|Energy|
> |:--:|:--:|:--:|:--:|
> |TimeGAN|176(min)|179(min)|217(min)|
> |Diffusion-TS|17(min)|15(min)|60(min)|
>
> > How well the performance of your model can be when there’s big jump in observed data?
>
> We thank the reviewer for this comment. We agree that further exploration/confirmation how well the performance the method can be under times series of real data with different unique features is a meaningful direction of work. We collected Google stock data from 2004 to 2022, and used data from 2004 to 2019 as the training set. Then we conducted predictions on the remaining data set (See Figure 11 in Appendix C.3). Although the accuracy of the results is relatively good, what we mainly discuss in this article is addressing the gap between time series generation (decomposition) and diffusion models. Thus we do not think this issue is the focus of our work and will treat it as a work in the near future.

---

### Meta-Review · Area_Chair_NAmM · 2023-12-09

**Metareview:**

The paper presents a novel transformer-based diffusion model, Diffusion-TS, for multivariate time series generation, which offers several innovative features. The model utilizes representations from existing work for trend and seasonality, directly estimates observations from arbitrary diffusion steps, and incorporates a Fourier-based loss term. Through comprehensive evaluations, Diffusion-TS outperforms existing generative models, demonstrating its efficacy in high-dimensional settings. Moreover, it readily extends to conditional generation for forecasting and imputation, solidifying its potential for diverse applications. Given these strengths, I think the paper will make a significant contribution to the field of time series generation and thus recommend for acceptance.

**Justification For Why Not Higher Score:**

While the proposed method shows promise for seasonal time series data, its performance and novelty need further validation and clarification. Addressing the reviewer's concerns would significantly improve the paper's strength and clarity.

**Justification For Why Not Lower Score:**

The paper presents a novel time series generation method, Diffusion-TS, which combines trend-seasonality decomposition with diffusion models. This unique approach outperforms existing methods in both unconditional and conditional generation tasks across various datasets. The ablation studies and qualitative results further demonstrate the effectiveness of the proposed method, solidifying its position as a significant contribution to the field of time series generation.

---

### Decision · Program_Chairs · 2024-01-16

Accept (poster)